# Molecule Relaxation by Reverse Diffusion with Time Step Prediction

## Abstract

Molecule relaxation—finding the stable state of an unstable configuration—is an important subtask for exploring the chemical compound space, for instance, to identify novel drugs or catalysts. Existing methods rely on local energy minimization with the gradients (i.e., force field) estimated through computationally intensive ab initio methods or approximated by a neural network trained on large expensive datasets encompassing *labeled stable and unstable* molecules. In this work, we propose molecule relaxation by reverse diffusion (MoreRed), a novel purely statistical approach where unstable molecules are seen as *noisy* samples to be denoised by a diffusion model equipped with a time step predictor to handle arbitrarily noisy inputs. Notably, MoreRed learns a simpler pseudo energy surface instead of the complex physical energy surface and is trained on a significantly smaller dataset consisting of solely *unlabeled stable* molecules, which is considerably less expensive to generate. Nevertheless, our experiments demonstrate its competitive performance to the state-of-the-art baseline in terms of the quality of the relaxed molecules inferred. Furthermore, we identify the high potential that time step prediction has to enhance the performance of data generation, where our findings are promising both in molecular structure and image generation.

## 1 Introduction

The exploration of chemical compound space is an important quest for quantum chemistry with applications to emerge in domains such as drug discovery, battery design, and catalyst development (Hajduk & Greer, 2007; Hautier et al., 2011; Bhowmik et al., 2019; Freeze et al., 2019; Gantzer et al., 2020; von Lilienfeld et al., 2020). Finding useful compounds requires to identify stable molecules with desirable properties, where *relaxation* plays a crucial role. I.e. given an unstable molecule, typically proposed by a computationally cheap method, the relaxation task adjusts the atomic positions in space so that the potential energy of the resulting molecular configuration reaches a local minimum – a procedure which is typically performed by gradient descent of the (physical) potential energy. Traditionally, the gradients to the quantum mechanical energy, which correspond to the forces, are computed using computationally demanding ab-initio atomistic simulation methods like density functional theory (DFT) (Hohenberg & Kohn, 1964), where the Schrödinger equation is approximately solved (Schrödinger, 1926). Therefore, this gradient computation process poses a serious bottleneck for exploring novel molecules and materials.

Over the past decade, machine learning approaches have emerged as promising alternatives. Force field (FF) models (Rupp et al., 2012; De et al., 2016; Chmiela et al., 2017; Schütt et al., 2017; Faber et al., 2018; Thomas et al., 2018; Noé et al., 2020; Unke et al., 2021b) based on kernel methods or neural networks can accurately learn the potential energy surface. They significantly accelerate the gradient descent optimization for molecule relaxation and thus facilitate chemical compound space exploration. Those approaches, however, have a drawback in the preparation of training datasets. Since the FF models need to predict the gradient at any point on any trajectory between any possible input molecule to its stable states, the training distribution must cover a wide range of the chemical space including stable and unstable molecules. Generating such training samples through DFT-based molecular dynamics simulations is expensive, and it is too time-consuming to apply it to large molecules. Therefore, established publicly available datasets such as QM7-X (Hoja et al., 2021) or ANI-1 (Smith et al., 2017a) contain only small structures with at most 8 heavy atoms and relatively few unique compounds.

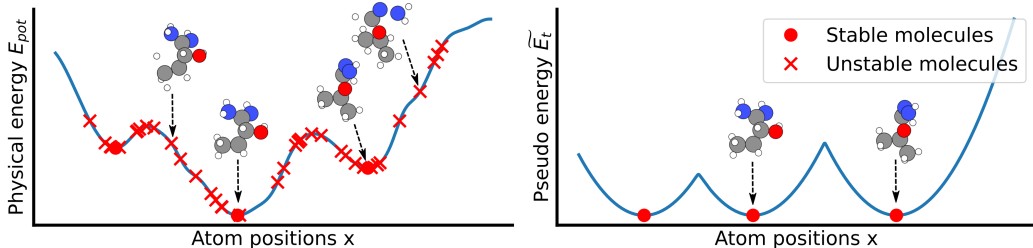

Figure 1: Schematic depictions of the physical potential energy (left) and the pseudo energy (right), which corresponds to the negative log of a mixture of isometric Gaussians with the common variance depending on the diffusion time step. Learning the physical energy requires stable (circles) and many unstable (crosses) training molecules, while learning the pseudo energy only requires stable training molecules.

In this work, we propose a novel statistical approach to molecule relaxation through reverse diffusion (MoreRed), where unstable input molecules are *denoised* by the reverse diffusion process. A notable difference from the existing methods is that MoreRed does not learn the intricate physical energy surface but a simple pseudo energy surface (see Figure 1). This brings a significant advantage over the FF models: Training MoreRed requires only unlabeled stable molecules, which considerably reduces the computational costs of generating training datasets. Therefore, it potentially expands the applicability of ML-driven relaxation to domains where force fields cannot be trained because only stable structures are reported. This comprises e.g. Materials Project (Jain et al., 2013) for crystal structure search or datasets with larger, more diverse organic molecules, as in PubChemQC (Nakata & Shimazaki, 2017) (up to 51 heavy atoms) or QMugs (Isert et al., 2022) (up to 100 heavy atoms). In our experiment on the QM7-X dataset, MoreRed, trained on two orders of magnitude less molecules, outperforms the state-of-the-art FF model. Furthermore, it exhibits increased robustness against the variation of the noise distribution of test structures, which increases the flexibility when choosing cheap methods to obtain candidates for relaxation.

A key component of MoreRed is the diffusion time step predictor, which enables to determine an appropriate diffusion time for an arbitrarily noisy input molecule. We provide a theoretical argument on why accurate prediction is possible. Beyond molecule relaxation, we explore the possibility of improving unconditional data generation performance by using an adaptive schedule based on the diffusion time predictor. Our experiments show promising results: Compared to the standard reverse diffusion with a predetermined schedule, our reverse diffusion with time step adaptation generates significantly more stable structures on the QM9 dataset (Ramakrishnan et al., 2014) and achieves lower FID scores on CIFAR-10 (Krizhevsky et al., 2009) image generation.

**Related Work:** A plethora of machine learning models that can accurately capture the physical potential energy surface have been proposed (Behler & Parrinello, 2007; Schütt et al., 2017; Chmiela et al., 2019; Smith et al., 2017b; Gasteiger et al., 2020; Batzner et al., 2022; Satorras et al., 2021; Frank et al., 2022; Batatia et al., 2022; Unke et al., 2022; Musaelian et al., 2023) and it has been shown that such approaches yield promising results in relaxation tasks (Unke & Meuwly, 2019; Unke et al., 2021a). However, as previously discussed, these models necessitate extensive datasets that encompass a substantial number of unstable molecules for training, which are costly to obtain. In order to improve stable structure property prediction based on unstable inputs, Uni-Mol+ Lu et al. (2023) learns to relax slightly perturbed structures resulting from a linear, noisy interpolation between stable and unstable systems. We also frame relaxation as a denoising problem, but our method is based on diffusion models and requires only stable molecules for training. Diffusion models are a class of generative models initially introduced by Sohl-Dickstein et al. (2015), where we adopt the standard denoising diffusion probabilistic models (DDPM) variant proposed by Ho et al. (2020). Diffusion models have previously been employed for stable molecule generation (Hoogeboom et al., 2022; Wu et al., 2022; Huang et al., 2023; Xu et al., 2023; Peng et al., 2023), conformation search (Xu et al., 2022), and molecular graph generation (Vignac et al., 2023; Kong et al., 2023) but they lack capabilities to start the reverse diffusion process from arbitrarily noisy samples. In MoreRed, we tackle this problem by predicting the diffusion time step.

There exist several other generative models for 3D molecular structures but none of them are designed for denoising. Typically, they generate stable molecules from scratch by iteratively adding new atoms (Gebauer et al., 2019; 2022; Simm et al., 2020; 2021; Meldgaard et al., 2022) or by transforming samples from a prior distribution (Noé et al., 2019; Köhler et al., 2020; Garcia Satorras et al., 2021; Klein et al., 2023). Furthermore, generative models have been used to sample conformations given molecular graphs (Mansimov et al., 2019; Simm & Hernandez-Lobato, 2020; Gogineni et al., 2020; Ganea et al., 2021; Xu et al., 2021; Lemm et al., 2021; Jing et al., 2022). Conventional conformer search algorithms systematically generate unstable candidate structures, e.g. with cheap empirical force fields, which are then optimized with molecule relaxation (Pracht et al., 2020).

## 2 BACKGROUND

### 2.1 MOLECULE RELAXATION

A molecule can be described concisely by the positions of its atoms in 3D space, denoted by the flattened positions vector $\mathbf{x} \in \mathbb{R}^{3N}$, and the types of its atoms, represented by their nuclear charges, denoted as $\boldsymbol{Z} \in \mathbb{N}^N$, where $N$ represents the number of atoms in the molecule. *Molecule relaxation* aims at the minimization of the physical potential energy of a molecule by adjusting the atom positions: $\min_{\mathbf{x}} E_{pot}(\mathbf{x}, \boldsymbol{Z})$. Given that the atom types $\boldsymbol{Z}$ remain constant throughout the optimization process, we will omit them from our notation for the remainder of this work. We refer to the optimized structures at local minima of the potential energy surface as *stable molecules* (and call all other configurations *unstable*). In many applications, the goal is to determine the properties of the stable structures, which makes molecule relaxation an integral part of the exploration of chemical compound space.

A standard approach to molecule relaxation is to adjust the atom positions using a local optimizer, e.g. L-BFGS (Fletcher, 2000; Liu & Nocedal, 1989), where the gradients $\frac{\partial E_{pot}}{\partial \mathbf{x}}$ are computed with ab initio methods like DFT (Parr & Yang, 1994) or wave function theory (Helgaker et al., 2013), which approximate a solution to the Schrödinger equation. This procedure is repeated until a convergence criterion is met, e.g. until the gradients are close to zero. Unfortunately, the ab initio methods scale at least cubically in the number of electrons of the molecule and are therefore very time-consuming. Consequently, it is common to use a cascade of methods with increasing computational demands to reduce the number of costly relaxation steps, where initial structures are obtained with cheap empirical force fields. However, even in this scenario, the ab initio calculations are prohibitive in cost. Machine learning force fields have been established as a tool to mitigate this problem. Once trained on a sufficiently diverse dataset of ab initio energies and forces of stable and unstable molecules, they can carry out the relaxation of the initial structures within a matter of seconds.

### 2.2 DIFFUSION MODELS

Diffusion models are a class of latent-variable generative models with a fixed encoding (forward) process and a trainable generative (reverse) process. It was originally proposed as DDPM (Sohl-Dickstein et al., 2015; Ho et al., 2020), and the close relation to score matching with Langevin dynamics (SMLD) (Song & Ermon, 2019; 2020) has been revealed in Song et al. (2021), where both methods are seen as discretized approximations to the continuous diffusion with the forward and reverse processes expressed as stochastic differential equations (SDE):

$$d\mathbf{x} = f(\mathbf{x}, t)dt + h(t)d\mathbf{w}, \qquad d\mathbf{x} = [f(\mathbf{x}, t) - h^2(t)\boldsymbol{\nabla}_{\mathbf{x}} \log p_t(\mathbf{x})]dt + h(t)d\bar{\mathbf{w}}, \qquad (1)$$

where $f(., t)$ and $h(t)$ are predefined functions describing the drift and diffusion coefficient, respectively, $\mathbf{w}$ and $\bar{\mathbf{w}}$ are the standard Wiener process and its time inversion and $p_t(\mathbf{x}_t) = \int p(\mathbf{x}_t|\mathbf{x}_0)p_{data}(\mathbf{x}_0)d\mathbf{x}_0$ is the diffused distribution at the time $t \in [0, T]$ with the diffusion kernel $p(\mathbf{x}_t|\mathbf{x}_0)$. Note that $p_0(\mathbf{x}_0) = p_{data}(\mathbf{x}_0)$ corresponds to the data distribution, while $p_T(\mathbf{x}_T)$ is the latent prior, e.g. $\mathcal{N}(\mathbf{x}_T; \mathbf{0}, \boldsymbol{I})$. The reverse diffusion process in Eq.(1) requires knowledge about the blurred data distribution through its derivative of log, i.e., $\boldsymbol{\nabla}_{\mathbf{x}} \log p_t(\mathbf{x})$, called the score function. In diffusion models, the score function is approximated by a neural network $\mathbf{s}_\theta(\mathbf{x}_t, t)$ trained by, e.g., denoising score matching (Vincent, 2011; Song & Ermon, 2019).

The standard variance-preserving DDPM (Ho et al., 2020), which we will adopt as our basic method, corresponds to a discretization of the SDE (1) with $f(\mathbf{x}, t) = -\frac{1}{2}\beta_t \mathbf{x}$ and $h(t) = \sqrt{\beta_t}$, where $\{\beta_t\}_{t=0}^{T}$ specifies the noise schedule. The forward and reverse diffusion updates are given as

$$\mathbf{x}_t = \sqrt{1 - \beta_t}\mathbf{x}_{t-1} + \sqrt{\beta_t}\boldsymbol{\varepsilon}, \quad \boldsymbol{\varepsilon} \sim \mathcal{N}(\mathbf{0}, \boldsymbol{I}), \tag{2}$$

$$\mathbf{x}_{t-1} = \frac{1}{\sqrt{1-\beta_t}}\left(\mathbf{x}_t + \frac{\beta_t}{\sqrt{1-\bar{\alpha}_t}}\boldsymbol{\varepsilon}_\theta(\mathbf{x}_t, t)\right) + \sqrt{\beta_t}\bar{\boldsymbol{\varepsilon}}, \quad \bar{\boldsymbol{\varepsilon}} \sim \mathcal{N}(\mathbf{0}, \boldsymbol{I}), \tag{3}$$

where $\bar{\alpha}_t = \prod_{s=1}^{t} \alpha_s$, $\alpha_t = 1 - \beta_t$, and the diffusion noise, which is proportional to the score function, is predicted by a neural network $\boldsymbol{\varepsilon}_\theta(\mathbf{x}_t, t) = -\sqrt{1-\bar{\alpha}_t}\mathbf{s}_\theta(\mathbf{x}_t, t)$ trained by minimizing:

$$L_{\text{DDPM}} = \mathbb{E}_{t \sim \mathcal{U}(1,T), \mathbf{x}_0 \sim p_{data}, \boldsymbol{\varepsilon} \sim \mathcal{N}(\mathbf{0}, \boldsymbol{I})}\left[||\boldsymbol{\varepsilon} - \boldsymbol{\varepsilon}_\theta(\mathbf{x}_t, t)||^2\right], \text{ where } \mathbf{x}_t = \sqrt{\bar{\alpha}_t}\mathbf{x}_0 + \sqrt{1-\bar{\alpha}_t}\boldsymbol{\varepsilon}. \tag{4}$$

A more comprehensive explanation of diffusion models is given in Appendix A.1.1

## 3 PROPOSED METHODS

In this work, we introduce a novel statistical approach to structure relaxation of molecules by framing it as a denoising problem, leveraging the capabilities of diffusion models. In the following, we present our method MoreRed and its key component, the diffusion time step predictor. Furthermore, we explain three different variants of MoreRed that we will evaluate in our experiments on molecule relaxation. Appendix A.2 provides further explanations with pseudo-code for training and sampling.

### 3.1 MoreRed: Molecule Relaxation by Reverse Diffusion

We propose to perform molecule relaxation by reverse diffusion (MoreRed), where the unstable molecules are seen as noisy versions of stable molecules, and relaxation is performed as denoising by the reverse diffusion process. Specifically, MoreRed learns the distribution of *stable* molecules, as the target data distribution $p_{data}(\mathbf{x}_0)$, and its Gaussian blurred versions:

$$p_t(\mathbf{x}_t) = \int p(\mathbf{x}_t|\mathbf{x}_0)p_{data}(\mathbf{x}_0)d\mathbf{x}_0, \qquad \text{for } t \in [0, T].$$

This amounts to learning the pseudo-energy, $\widetilde{E}_t = -\log p_t(\mathbf{x})$, which is much simpler than the physical potential energy $E_{pot}(\mathbf{x})$ that the existing FF models need to learn (see Figure 1). More importantly, this drastically reduces the cost of preparing the training data. For making FF models reliable at any possible input molecule, the training data needs many unstable molecules. These are usually obtained by exploring the chemical space close to stable structures by approximate sampling from the Boltzmann distribution $p(\mathbf{x}) \propto \exp(-\beta E_{pot}(\mathbf{x}))$ at a sufficiently large finite temperature, as for example with molecular dynamics. In contrast, the diffusion model only requires stable molecules as training data and the whole data space is covered simply by Gaussian blurring in the forward diffusion process. We adopt the standard DDPM framework, of which the diffusion updates and the training loss are given in Eqs. (2)–(4).

For statistical efficiency, we incorporate the physical symmetry of molecules. Namely, we design our diffusion model so that the distribution is invariant, i.e., $p_t(\mathcal{T}(\mathcal{R}(\mathbf{x}_t))) = p_t(\mathbf{x}_t)$, for any rotation $\mathcal{R}(\cdot)$ and translation $\mathcal{T}(\cdot)$ operations. We use an equivariant noise model $\boldsymbol{\varepsilon}_\theta(\mathbf{x}_t, t)$ and rotation invariant latent prior $\mathcal{N}(\mathbf{0}, \boldsymbol{I})$, which guarantee the rotational invariance (Xu et al., 2022). For the equivariant noise model, we adopt the equivariant message passing architecture PaiNN (Schütt et al., 2021). It allows us to directly predict rotational equivariant tensor properties, such as diffusion noise, as well as invariant scalar properties. To guarantee the translational invariance, we enforce a zero centre of geometry (CoG) of molecules by centring the samples from the latent prior (Köhler et al., 2020) and the diffusion noise $\boldsymbol{\varepsilon}_t$ as well as the predicted noise $\boldsymbol{\varepsilon}_\theta(\mathbf{x}_t, t)$ at every step (Xu et al., 2022; Hoogeboom et al., 2022).

To perform reverse diffusion starting from unstable molecules at arbitrary noise levels (i.e. not sampled from the noise distribution $p_T(\mathbf{x}_T)$), it is necessary to set the initial diffusion time step appropriately. More specifically, we need to know from which diffused distribution $p_t$, the input molecule $\tilde{\mathbf{x}}$ was drawn. It is necessary to estimate an appropriate diffusion time step $\hat{t}$ such that $p_{\hat{t}}(\tilde{\mathbf{x}})$ is sufficiently high. Therefore, we propose a time step predictor in the next subsection.

## 3.2 DIFFUSION TIME STEP PREDICTION

In order to identify the noise level of unstable molecules that we want to relax, we train a model $g_\phi(\mathbf{x}_t)$ to predict the diffusion time step by minimizing

$$L_{\text{DPT}} = \mathbb{E}_{t \sim \mathcal{U}(1,T), \mathbf{x}_0 \sim q(\mathbf{x}_0), \boldsymbol{\varepsilon} \sim \mathcal{N}(\mathbf{0}, \boldsymbol{I})} \left[ (g_\phi(\mathbf{x}_t) - a(t))^2 \right], \tag{5}$$

where $a(t)$ is a monotonic function to scale the output, e.g., $a(t) = t/T$. For $g_\phi(\mathbf{x}_t)$, we again adopt the neural network architecture PaiNN (Schütt et al., 2021) but use the scalar features to predict the time step as it is an invariant quantity.

Considering that stable molecules are isolated from each other up to the symmetry operations, our target distribution $p_t$ is essentially a mixture of Gaussians with each component centered at a training molecule and the variance monotonically increasing with the diffusion time step $t$. Therefore, the time step prediction amounts to noise level estimation, which is not difficult when the dimensionality $D$ is large and the diffusion noise $\sigma_t$ is small for the following reasons. As discussed in Bishop (2006), in the polar coordinate system with the radius $r \geq 0$, the marginal distribution of the Gaussian in a scaled radius $\tilde{r} = \frac{r}{\sqrt{D}\sigma}$ is given as

$$p(\tilde{r}) = \frac{D^{D/2} \tilde{r}^{D-1}}{2^{D/2-1} \Gamma(D/2)} \exp \left( -\frac{D\tilde{r}^2}{2} \right), \tag{6}$$

where $\Gamma(\cdot)$ denotes the Gamma function. For large $D$, this has a sharp peak around $\hat{r} \approx 1$, implying that the density is concentrated in a thin shell (see Figure A1 left). Accordingly, when the noise level $\sigma_t$ is small so that the mixture components do not overlap, most of the diffused samples $\mathbf{x}_t$ have similar noise levels, which is easy to identify. This is similar to the fact that the noise level can be easily recognized in noisy images with a sufficiently large number of pixels. When the noise level increases, the diffused samples from different training samples overlap with each other, making the diffusion time step estimation more difficult. Our evaluation of the diffusion time step predictor in Appendix A.4.2 matches the intuition described above. Note that, due to the high dimensionality of the input space, the mixture overlap does not significantly affect the prediction performance for samples with small noise, as explained in Appendix A.2.1 along with the derivation of Eq.(6).

## 3.3 VARIANTS OF REVERSE DIFFUSION

In our molecule relaxation experiments, we compare three variants of MoreRed that differ in how they handle the diffusion time step prediction. In the first variant, called *MoreRed initial time prediction* (MoreRed-ITP), only the initial diffusion time step is predicted. This means that given an unstable input molecule $\tilde{\mathbf{x}}$, it estimates an appropriate starting diffusion time step, i.e., $\hat{t} = g_\phi(\tilde{\mathbf{x}})$, sets $\mathbf{x}_{\hat{t}} = \tilde{\mathbf{x}}$, and performs the reverse diffusion update using the fixed standard DDPM reverse process (3) for $t = \hat{t}, \hat{t} - 1, \ldots, 0$.

As a second variant, we use a more flexible process where the time step prediction is performed at every denoising step. We call this approach *MoreRed adaptive scheduling* (MoreRed-AS). It iterates through a time-adaptive version of Eq.(3):

$$\hat{t} \leftarrow g_\phi(\tilde{\mathbf{x}}), \qquad x_{\hat{t}-1} \leftarrow \frac{1}{\sqrt{1-\beta_{\hat{t}}}} \left( \mathbf{x}_{\hat{t}} + \frac{\beta_{\hat{t}}}{\sqrt{1-\bar{\alpha}_{\hat{t}}}} \boldsymbol{\varepsilon}_\theta(\mathbf{x}_{\hat{t}}, \hat{t}) \right) + \sqrt{\beta_{\hat{t}}} \bar{\varepsilon}, \quad \bar{\varepsilon} \sim \mathcal{N}(\mathbf{0}, \boldsymbol{I}). \tag{7}$$

In contrast to the fixed schedule, this allows the reverse process to skip time steps if the current sample converges faster, or to increase the time step to correct for errors in the denoising model. Similar to classical molecular relaxation methods, we define a convergence criterion for stopping the denoising process, i.e. we require time step predictions smaller than a threshold $\hat{t} \leq \underline{t}$. The idea of adaptive scheduling does not specifically target the molecule relaxation task but also benefits diffusion models in general, because errors due to inaccurate noise predictions or the stochasticity of the denoiser can accumulate, and therefore using a schedule with a fixed number of steps can lead to samples that deviate from the optimal reverse trajectory (Song et al., 2021). We explore this potential in Section 4.3.

A third variant, *MoreRed joint training* (MoreRed-JT), uses the same adaptive reverse diffusion process (7) as MoreRed-AS but instead of training two separate models, a single architecture predicts both noise $\boldsymbol{\varepsilon}_\theta$ and the time step $g_\theta$. We train this network by minimizing the joint loss $L_{\text{joint}} = \eta L_{\text{DDPM}} + (1 - \eta) L_{\text{DTP}}$, for $\eta \in [0, 1]$ defining a trade-off between the two losses.

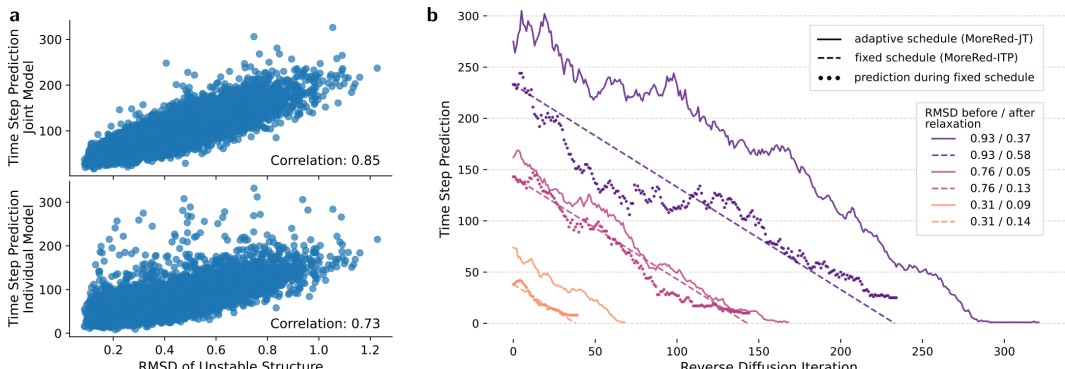

Figure 2: **a**: Scatter plots of the RMSD of 10k unstable test structures from QM7-X vs. the initial diffusion time step predicted by MoreRed. Top: The model that was jointly trained on denoising and time step prediction (MoreRed-JT). Bottom: The model that was individually trained on time step prediction (MoreRed-ITP/-AS). **b**: Comparison of time step trajectories for relaxation of three unstable test structures from QM7-X following an adaptive schedule (predicted with MoreRed-JT, solid lines) and a fixed schedule (MoreRed-ITP, dashed lines). The dots show the predicted (but not utilized) time step predictions for MoreRed-ITP during denoising with the fixed schedule.

## 4 EXPERIMENTS

Our MoreRed with the diffusion time predictor exhibits high performance in several experiments, see below. Note that MoreRed is based on the state-of-the-art equivariant neural network architecture PaiNN which we will also compare to as baseline for relaxation. Afterwards, we demonstrate that the adaptive diffusion scheduling with the diffusion time step predictor also enhances unconditional data generation, i.e., reverse diffusion from random noise, in molecular structure and image generation. We will make our code publicly accessible.[1]

### 4.1 DIFFUSION TIME STEP PREDICTION PERFORMANCE

The diffusion time step predictor is an integral part of MoreRed. It determines the starting step $\hat{t}$ of the reverse diffusion for relaxation of unstable molecules. Figure 2a shows that there is a clear correlation between the distance of unstable test molecules to their stable state and the predicted time step. We observe that jointly training the time step predictor with the noise predictor (top) leads to less outliers in the predictions of $\hat{t}$ and a higher correlation than training them separately (bottom). Besides, both variants reliably predict $\hat{t} > 0$ for all the 10k unstable structures, highlighting their robustness in identifying unstable structures even if these do not contain Gaussian noise.

In our adaptive variants MoreRed-AS/-JT, we predict $\hat{t}$ at every reverse diffusion iteration to follow the time-adaptive procedure described in Eq. 7. Figure 2b shows the merit of this approach. When following a fixed schedule (dashed lines) with one denoising step per reverse step, errors can occur and accumulate (Song et al., 2021). If not corrected, they lead to a mismatch between the true noise level in the structure and the time step $t$. Therefore, the reverse process may end before the sample reaches the data manifold, as evident from the high predicted time step values (depicted as dots) at the end of relaxation. The adaptive schedule (solid lines) can correct for such errors by adapting $\hat{t}$ at each reverse diffusion iteration. It successfully converges to $\hat{t} = 0$ and ends with molecules that are significantly closer to the ground truth stable state than the ones obtained with the fixed schedule (see the RMSD values after relaxation in the legend of Figure 2b).

### 4.2 MOLECULE RELAXATION PERFORMANCE

**Dataset:** We use QM7-X (Hoja et al., 2021), a comprehensive dataset of 4.2M labeled molecules that were derived from 7k molecular graphs sampled from the GDB13 chemical space with up to 7 heavy atoms, including types C, N, O, S, and Cl. For each stable molecule, there are 100

---

[1](providedlink).

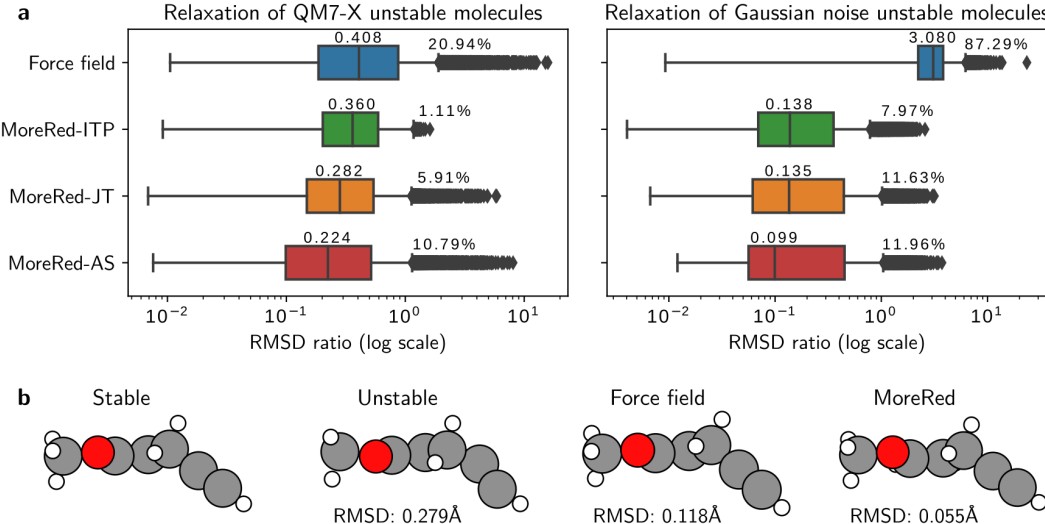

Figure 3: **a:** The RMSD ratios of molecules relaxed with the baseline FF model and with our MoreRed variants for 20k unstable structures from the QM7-X test split (left) and $7k$ structures obtained by adding Gaussian noise with 250 diffusion forward steps to stable structures from the QM7-X test split (right). The median values and the percentage of failure cases, i.e., the cases where the RMSD ratio exceeds 1, are shown above each box plot. **b:** An example result of molecule relaxation: A stable test molecule from QM7-X, a corresponding unstable input structure from the dataset, the molecule relaxed by the baseline FF model, and by our MoreRed-AS , from left to right, respectively.

corresponding unstable structures. Further details on the data are provided in Appendix A.4.1.

**Model architectures:** For a fair comparison, all variants of MoreRed, as well as the baseline force field (FF) model, use the equivariant message passing neural network PaiNN (Schütt et al., 2021) as a state-of-the-art backbone molecular representation. We utilize the implementation from the open source software package SchNetPack (Schütt et al., 2023). MoreRed is trained with a maximum diffusion time step of $T = 1000$, and we employ the polynomial approximation of the cosine noise schedule (Hoogeboom et al., 2022) with a precision parameter of $s = 10^{-5}$ to prevent the atoms from undergoing large unrealistic movements during the initial sampling steps. The baseline FF achieves an MAE of 16.31meV for energy and 22.52meV/Å for forces after training, which are on par with what Unke et al. (2021a) reported. Relaxation with the baseline FF is performed with the L-BFGS optimization algorithm implemented in ASE (Larsen et al., 2017), with a maximum number of 1000 iterations and the force magnitude convergence threshold of 5meV/Å. Further details on all neural network models including the hyperparameter settings are given in Appendix A.3.

**Results:** We evaluate the molecule relaxation performance of MoreRed, and compare it with the baseline FF model. As test inputs $\tilde{x}$, we choose 3 unstable conformers for each stable structure in the test split, resulting in around 20k unstable structures in total. Specifically, we sort the unstable molecules based on the root mean squared deviation (RMSD) from their stable structure, and choose 3 molecules equally spaced throughout the sorted list. This approach allows us to cover a wide range of test examples, from barely noisy with low RMSD values to noisier molecular structures with higher RMSD values. Note that, whenever we compute the RMSD, the rotation and the translation of molecules are aligned. Besides, we set a maximum number of relaxation steps equal to 1000 for all FF and MoreRed models and a convergence criterion of $\hat{t} \leq 0$ for MoreRed.

Figure 3a (left) shows the RMSD ratio, i.e., the RMSD after relaxation divided by the RMSD of the unstable initial structure, for the baseline FF model, and our three methods, MoreRed-ITP, MoreRed-JT, and MoreRed-AS. Remarkably, all our MoreRed variants outperform the baseline FF model, despite only being trained on stable molecules and therefore reducing the number of training points by a factor of 100. We define *failure cases* as the cases where the RMSD ratio exceeds 1, which means that the RMSD of the input structure was increased during molecule relaxation, i.e., the structure diverged from its stable target structure during the denoising process. Those failures correspond to the

cases where either optimization fails or the relaxation converges to a different local minimum in the potential energy surface. The percentage of the failure cases is shown upper right to each box plot. Compared to the FF model, our methods, especially, MoreRed-ITP and MoreRed-JT, drastically reduce the failure cases, clearly indicating high robustness. We provide an extended comparison of our three variants and additional results in Appendix A.4.3 and Figure A6. Additionally, to reinforce the robustness and generalizability of MoreRed, we run the same experiment using SO3Net (Schütt et al., 2023) as an alternative equivariant representation to PAiNN. As illustrated in Figure A7, MoreRed still outperforms the FF. We refer to Appendix A.4.4 for more details. Figure 3b illustrates an example of molecule relaxation, where the unstable input molecule, the ground-truth stable state, and the relaxed molecules by the FF model and MoreRed-AS are shown. We see that the hydrogen atoms (white) exhibit a pronounced displacement in the unstable input state, resulting in RMSD= 0.279Å, and that the relaxed molecules are similar to the ground-truth, while quantitatively, MoreRed-AS gives a lower RMSD =0.055Å than the FF model giving RMSD = 0.118Å.

We also evaluate the methods on synthetically generated test inputs by diffusing stable test molecules for 250 forward diffusion steps. We ensure that the resulting median RMSD of the diffused configurations matches the median RMSD of the unstable molecules from the test split of QM7-X. Figure 3a (right) shows the RMSD ratios after molecular relaxation. We observe that the baseline FF fails to optimize the noisy structures in almost 90% of the cases. This is because synthetic test inputs can fall out of the QM7-X training distribution (that follows the Boltzmann distribution). On the other hand, MoreRed performs well in both tasks, on the synthetic samples and on the QM7-X test split samples. This is because it is trained on diffused molecules, which cover a larger part of chemical space, albeit requiring less training data points. This allows MoreRed to be used for molecule relaxation of input structures that are obtained from different sources, e.g. different datasets, various empirical force fields, or other generative models, while still yielding a robust performance.

Training MoreRed models until complete convergence takes 35 hours on average, while the force field model needs more than 7 days on NVIDIA P100. However, one inference step with a batch size of 128 during denoising takes 0.03s for the force field model against 0.05s for MoreRed. The median number of relaxation steps until convergence is 118 for the FF model, 53, 219, and 1000 for MoreRed-ITP, MoreRed-JT, and MoreRed-AS, respectively in the test split experiment (Figure 3a left). A more detailed comparison of computation times can be found in Appendix A.4.5.

### 4.3 DATA GENERATION WITH DIFFUSION TIME STEP PREDICTION

In contrast to existing diffusion models, MoreRed-AS/-JT adaptively control the reverse diffusion process with the diffusion time step predictor, as observed in the relaxation task (see Figure 2b). In the following, we demonstrate the potential of this idea to enhance data generation performance for both molecules and images. For a fair comparison, we employ the same diffusion model for classical sampling with a fixed schedule (DDPM) and for adaptive sampling using the time step predictor (MoreRed). In this section, we refer to MoreRed-AS as MoreRed. In Appendix A.4.7, we add figures with examples of generated samples.

**Molecular Structure Generation:** We evaluate the molecule structure generation performance on the QM9 dataset (Ramakrishnan et al., 2014), a widely used benchmark for the molecular generation tasks. It comprises approximately 130k stable organic molecules, each containing up to 9 heavy atoms of types C, O, N, and F. We use 55k molecules for training, 10k for validation, e.g. for scheduling the learning rate, and define the rest as the test split. For data generation with MoreRed, we set the convergence criteria to $\hat{t} \leq 0$ and the maximum number of steps to 2000. In the standard DDPM (Ho et al., 2020), we use a fixed schedule with $T = 1000$ (same as during training). For our evaluation, we generate 10k structures from the latent prior distribution, where we use the atomic compositions $Z$ of molecules randomly drawn from the test split.

As metrics, we adopt validity, uniqueness and novelty as proposed by Gebauer et al. (2019), using their publicly available analysis script for comparability. It translates the generated structures to canonical SMILES (Weininger, 1988) encodings, which is a string representation of molecular graphs. A molecule is considered valid if all its atoms are connected and possess the proper valency in that encoding. Furthermore, unique and novel molecules are identified by comparing the canonical SMILES strings of all generated structures to each other and to those of all molecules in QM9, respectively. Table 1 summarizes the results, where "+Unique" indicates the proportion of

Table 1: Quality of 10k generated molecules after training on QM9. DDPM-large and MoreRed-large use 4 times more parameters as larger architecture.

| Model | Valid (%) | +Unique (%) | +Novel (%) |
|---|---|---|---|
| DDPM | 78.2 | 77.3 | 62.5 |
| MoreRed | **89.3** | **88.0** | **68.6** |
| DDPM-large | 86.6 | 85.3 | 63.8 |
| MoreRed-large | **94.7** | **92.4** | **66.7** |

Table 2: FID scores of unconditioned image generation on CIFAR-10 with linear and cosine noise schedules.

| Model | Schedule | FID |
|---|---|---|
| DDPM | Linear | 5.66 |
| MoreRed | Linear | **5.14** |
| DDPM | Cosine | 28.97 |
| MoreRed | Cosine | **6.80** |

generated molecules that are stable and unique, and "+Novel" indicates the proportion of molecules that are stable, unique, and novel. We observe that MoreRed, i.e. adaptive scheduling, performs better than DDPM, i.e. fixed scheduling, in all criteria. The same tendency can also be observed for architectures with more parameters, i.e. MoreRed-large. This confirms our hypothesis that the adaptive reverse diffusion procedure based on the time step prediction is beneficial for unconditional sampling from Gaussian noise. MoreRed can adapt the time step, as can be seen in some exemplary sampling trajectories in Figure A8, to correct for errors. We further discuss these findings in Appendix A.4.5, where we also report an extended comparison to other generative models in Table A1.

**Image Generation:** To further validate the generalizability of our approach to different data modalities, we employ the UNet architecture to evaluate unconditioned image generation performance on CIFAR-10 (Krizhevsky et al., 2009). Details on the architecture and hyperparameters are found in Appendix A.3.2. Similar to molecular structure generation, we set the convergence criteria to $\hat{t} \leq 0$, the maximum number of steps to 2000 and $T = 1000$. Table 2 shows the Fréchet inception distance (FID) score (Heusel et al., 2017) for the baseline DDPM and our MoreRed with linear and cosine noise schedules. MoreRed again outperforms DDPM, demonstrating the potential of the diffusion time predictor to enhance data generation—applying it to longer diffusion with larger architectures could benefit the state-of-the-art.

## 5 CONCLUSION

Diffusion processes (Jacobs, 1935; Ikeda & Watanabe, 1989) allow to accurately model complex high-dimensional stochastic dependencies and have in recent years – due to their high efficiency, accuracy and conceptual elegance – gained high popularity in computer vision and beyond. Reversing diffusion has become a workhorse for generation, denoising, inpainting, interpolation, solving inverse problems, etc. (Sohl-Dickstein et al., 2015; Ho et al., 2020; Song & Ermon, 2019; Chung et al., 2023; Su et al., 2022).

In this work we contribute two novel aspects to this interesting and active research direction: (1) we establish three novel reverse diffusion models with time step prediction (MoreRed ITP, JT, AS), and (2) show their immediate usefulness in a challenging application from quantum chemistry, namely molecule relaxation, and in molecule and image generation. Molecule relaxation is an important, ubiquitous and highly time-consuming task when exploring the chemical compound space (von Lilienfeld et al., 2020). Here, instead of relying on density functional theory, we employ MoreRed, with the idea of *statistically denoising unstable molecules*. The result of this *denoising* process is finding the relaxed or stable molecule while avoiding the high computational load of an ab-initio simulation for relaxation at the same accuracy. Notably, MoreRed is able to learn with surprisingly high accuracy by effectively reconstructing a simpler pseudo-energy surface instead of the complex physical energy surface from a dataset consisting of solely *unlabeled stable* molecules. This is not only considerably less expensive to generate but requires also fewer samples than state-of-the-art machine learning force field models – at the same time comparing favourably in terms of the quality of the resulting relaxed molecules. In addition, MoreRed impressively shows the usefulness of adaptive schedules with time step prediction in reverse diffusion for enhancing data generation, with promise both for molecular structure and image generation.

Future work will study explanation methods (XAI) to extract the unknown strategies of the reverse model. Furthermore, we will aim to scale the proposed models for larger molecules and materials.

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

# A  APPENDIX

## A.1  DETAILED BACKGROUND

### A.1.1  DIFFUSION MODELS

Diffusion models are a class of latent-variable generative models that learn to reverse a diffusion process by estimating small perturbations in the data, enabling sampling via an iterative transition from a tractable noise distribution to a complex data distribution. These models encompass a predefined forward process or inference model $q$, which progressively adds noise to destroy the signal in the target data $\mathbf{x}_0$ over $T$ steps $\mathbf{x}_1, \mathbf{x}_2, \ldots, \mathbf{x}_T$, and a learned backward process or generative model $p_\theta$, responsible for reversing the forward process by progressively eliminating the noise starting from $x_T$ to reconstruct the original signal (Sohl-Dickstein et al., 2015).

By starting from the complex data distribution of interest $q(\mathbf{x}_0)$ and fixing the forward process to the Gaussian Markovian kernel $q(\mathbf{x}_t|\mathbf{x}_{t-1}) = \mathcal{N}(\mathbf{x}_t; \sqrt{1-\beta_t}\mathbf{x}_{t-1}, \beta_t \boldsymbol{I})$ with a noise schedule $\{\beta_t \in (0,1)\}_{t=0}^T$, we can derive, in closed-form, the marginal distribution $q(\mathbf{x}_t|\mathbf{x}_0) = \mathcal{N}(\mathbf{x}_t; \sqrt{\bar{\alpha}_t}\mathbf{x}_0, (1-\bar{\alpha}_t)\boldsymbol{I})$ at any diffusion step $t$, where $\bar{\alpha}_t = \prod_{s=1}^t \alpha_s$ and $\alpha_t = 1 - \beta_t$ (Ho et al., 2020). As $T \to \infty$, the endpoint variable $\mathbf{x}_T$ converges towards the tractable isotropic Gaussian distribution, i.e. $q(\mathbf{x}_T|\mathbf{x}_0) \approx \mathcal{N}(\mathbf{x}_T; \mathbf{0}, \boldsymbol{I})$. Intriguingly, efficient sampling at any step $t$ is ensured through the Gaussian reparametrization trick: $\mathbf{x}_t = \sqrt{\bar{\alpha}_t}\mathbf{x}_0 + \sqrt{1-\bar{\alpha}_t}\varepsilon$, with $\varepsilon \sim \mathcal{N}(\mathbf{0}, \boldsymbol{I})$.

The backward process $p$ is the reverse generative process of interest when sampling from the target distribution, starting from $p(\mathbf{x}_T) = \mathcal{N}(\mathbf{x}_T; \mathbf{0}, \boldsymbol{I})$. While computing the reverse chain $q(\mathbf{x}_{t-1}|\mathbf{x}_t)$ is intractable because we need to integrate over all the data, the forward and the reverse process share the same Gaussian functional form, provided $\beta_t$ is sufficiently small, i.e. $T$ is large (Feller, 2015). Furthermore, the forward process posteriors $q(\mathbf{x}_{t-1}|\mathbf{x}_t, \mathbf{x}_0)$ are tractable when conditioned on $\mathbf{x}_0$. Therefore, to approximate these posteriors, Ho et al. (2020) introduced the noise parametrization $q(\mathbf{x}_{t-1}|\mathbf{x}_t, \mathbf{x}_0) = p_\theta(\mathbf{x}_{t-1}|\mathbf{x}_t) = \mathcal{N}(\mathbf{x}_{t-1}; \boldsymbol{\mu}_\theta(\mathbf{x}_t, t), \sigma_t \boldsymbol{I})$, with $\boldsymbol{\mu}_\theta(\mathbf{x}_t, t) = \frac{1}{\sqrt{\alpha_t}}\left(\mathbf{x}_t - \frac{1-\alpha_t}{\sqrt{1-\bar{\alpha}_t}}\epsilon_\theta(\mathbf{x}_t, t)\right)$ and $\sigma_t = \frac{1-\bar{\alpha}_{t-1}}{1-\bar{\alpha}_t}\beta_t$. The latter parametrization allows efficient training of the noise model $\epsilon_\theta(\mathbf{x}_t, t)$ using the following simple objective, which is a reweighted version of the original variational lower bound:

$$L_{simple} = \mathbb{E}_{t\sim\mathcal{U}(1,T), \mathbf{x}_0\sim q(\mathbf{x}_0), \boldsymbol{\varepsilon}\sim\mathcal{N}(\mathbf{0},\boldsymbol{I})}\left[||\boldsymbol{\varepsilon} - \boldsymbol{\varepsilon}_\theta(\sqrt{\bar{\alpha}_t}\mathbf{x}_0 + \sqrt{1-\bar{\alpha}_t}\boldsymbol{\varepsilon}, t)||^2\right], \quad (8)$$

In this framework, learning reduces to predicting the Gaussian noise $\boldsymbol{\varepsilon}_\theta(\mathbf{x}_t, t)$ required to remove from each sample $\mathbf{x}_t$ at step $t$, thereby guiding $\mathbf{x}_{t-1}$ towards a sample $\mathbf{x}_0$ from the target distribution.

### A.1.2  RELATION TO SCORE-BASED MODELS

While developed independently, score-based and denoising diffusion models share similar methodologies. Score-based models (Song & Ermon, 2019; 2020) employ score-matching techniques (Hyvärinen & Dayan, 2005; Vincent, 2011; Song & Ermon, 2019; Song et al., 2020) to learn score functions from progressively noisier versions of the target distribution. Once trained, they utilize gradient-based Markov chain Monte Carlo (MCMC) techniques, such as Langevin dynamics (Parisi, 1981; Grenander & Miller, 1994), to iteratively draw samples, starting from the noisiest distribution and progressing to the cleanest. Interestingly, the noise model introduced in the earlier parametrization of diffusion models can be linked to the score function in score-based models through the equation $\boldsymbol{\varepsilon}_\theta(\mathbf{x}_t, t) = -\sqrt{1-\bar{\alpha}_t}\mathbf{s}_\theta(\mathbf{x}_t, t)$, with $\mathbf{s}_\theta(\mathbf{x}_t, t) = \nabla_{\mathbf{x}_t} \log p_\theta(\mathbf{x}_t) \approx \nabla_{\mathbf{x}_t} \log q(\mathbf{x}_t)$ being the score function (Song et al., 2021). The marginals $q(\mathbf{x}_t) = \int q(\mathbf{x}_t|\mathbf{x}_0)q(\mathbf{x}_0)d\mathbf{x}_0$ correspond to smoother versions of the target data distribution $\mathbf{x}_0$ with increasing $t$. Thus, diffusion models effectively learn the score functions of increasingly noisy versions of the target distribution. Despite slight differences in how noise is incorporated between diffusion and score-based models, the $L_{simple}$ 8 objective of diffusion models is identical to a weighted sum of denoising score-matching objectives (Vincent, 2011; Song & Ermon, 2019; Song et al., 2021). Consequently, this connection allows us to train a noise model $\boldsymbol{\varepsilon}_\theta(\mathbf{x}_t, t)$ using diffusion models and then recover the score model $\mathbf{s}_\theta(\mathbf{x}_t, t)$ using the equation mentioned earlier, which, in turn, enables the application of various score-based sampling techniques with diffusion models and vice versa, enhancing the flexibility and utility of both approaches.

### A.1.3 SAMPLING

Score-based models employ an annealed version of Langevin MCMC to sample sequentially from the marginals $p_\theta(\mathbf{x}_t) \approx q(\mathbf{x}_t)$ using the iterative equation:

$$\mathbf{x}_t^k = \mathbf{x}_t^{k-1} + \gamma_t \mathbf{s}_\theta(\mathbf{x}_t^{k-1}, t) + \sqrt{2\gamma_t}\mathbf{z}_t^k, \quad k = 1, 2, \ldots, K, \tag{9}$$

where $\mathbf{z}_t^k \sim \mathcal{N}(\mathbf{0}, \mathbf{I})$, and $\gamma_1 > 0$ is a damping coefficient to be tuned together with $K$. We run this procedure sequentially from $t = T$ down to $t = 1$ with $\mathbf{x}_T^0 \sim p(\mathbf{x}_T)$ and $\mathbf{x}_t^0 = \mathbf{x}_{t+1}^K$, assuming $q(\mathbf{x}_1) \approx q(\mathbf{x}_0)$ for negligible noise at $t = 1$. As $\gamma_t \to 0$ and $K \to \infty$, Langevin MCMC converges to a sample from $q(\mathbf{x}_t)$ (Song & Ermon, 2019; Song et al., 2021). In contrast, diffusion models use the estimated reverse transition probabilities $p_\theta(\mathbf{x}_{t-1}|\mathbf{x}_t)$ to perform ancestral sampling. They start with a sample from the prior $p(\mathbf{x}_T)$ and then sequentially apply the estimated reverse Markov chain to map $x_T$ to a target distribution sample $x_0$ using the iterative equation:

$$\mathbf{x}_{t-1} = \frac{1}{\sqrt{1-\beta_t}}\big(\mathbf{x}_t + \beta_t \mathbf{s}_\theta(\mathbf{x}_t, t)\big) + \sqrt{\beta_t}\mathbf{z}_t, \quad t = T, T-1, \ldots, 1, \tag{10}$$

(Ho et al., 2020). This process resembles annealed Langevin dynamics but with distinctions. As the backward process in diffusion models estimates the exact inverse of the forward one, the mean term is scaled by $(\sqrt{1-\beta_t})^{-1}$ due to the variance-preserving property. Denoising score matching, on the other hand, employs variance-exploding noise addition (Song et al., 2021). Additionally, the number of sampling iterations in diffusion models is typically equal to the diffusion steps in the forward process, with scaling factors and damping rates explicitly specified by the noise schedule. In contrast, score-based models offer flexibility with freely tuned hyperparameters $\gamma_t$ and $K$, allowing more than one sampling step per latent variable $\mathbf{x}_t$ to trade-off sample quality with computational cost.

Although sampling from diffusion models may seem more efficient and precisely define the sampling parameters, Song et al. (2021) demonstrated the effectiveness of a two-step predictor-corrector sampling method. In this approach, the predictor estimates the subsequent ancestral sample $\mathbf{x}_{t-1}$, using the diffusion model sampling formula, and the corrector applies Langevin dynamics to $\mathbf{x}_{t-1}$ for $K$ steps with a damping factor $\gamma$ to refine and correct the prediction, leading to improved sample quality already with $K = 1$. Thus, combining diffusion and score-based models during sampling introduces flexibility and a trade-off between sample quality and computational cost. While the predictor-corrector sampler mitigates sampling errors caused by the stochasticity of a single prediction step in diffusion models, the optimal $K$ and $\gamma$ for correction steps can vary between iterations and samples, making manual tuning suboptimal. Motivated by this concept, we propose a more efficient and adaptive alternative, i.e. the prediction of the diffusion step, to address these challenges.

### A.2 MORERED: DETAILS

### A.2.1 DIFFUSION TIME STEP PREDICTION

**Derivation of Eq.(6):** In the polar coordinate system, the marginal distribution of the centered isotropic Gaussian with respect to the radius is given by

$$p(r) = \frac{1}{(2\pi\sigma^2)^{D/2}} \exp\left(-\frac{r^2}{2\sigma^2}\right) \cdot r^{D-1}S_D$$
$$= \frac{r^{D-1}}{2^{D/2-1}\Gamma(D/2)\sigma^D} \exp\left(-\frac{r^2}{2\sigma^2}\right),$$

where $S_D = \frac{2\pi^{D/2}}{\Gamma(D/2)}$ is the surface area of the $((D-1)$-dimensional) unit sphere embedded in the $D$-dimensional space, and $\Gamma(\cdot)$ denotes the Gamma function (Bishop, 2006). By changing the variable from the radius to its scaled version, $\tilde{r} = \frac{r}{\sqrt{D}\sigma}$, we have

$$p(\tilde{r}) = p(r)\frac{dr}{d\tilde{r}} = \frac{(\sqrt{D}\sigma\tilde{r})^{D-1}}{2^{D/2-1}\Gamma(D/2)\sigma^D} \exp\left(-\frac{(\sqrt{D}\sigma\tilde{r})^2}{2\sigma^2}\right) \cdot \sqrt{D}\sigma$$
$$= \frac{D^{D/2}\tilde{r}^{D-1}}{2^{D/2-1}\Gamma(D/2)} \exp\left(-\frac{D\tilde{r}^2}{2}\right),$$

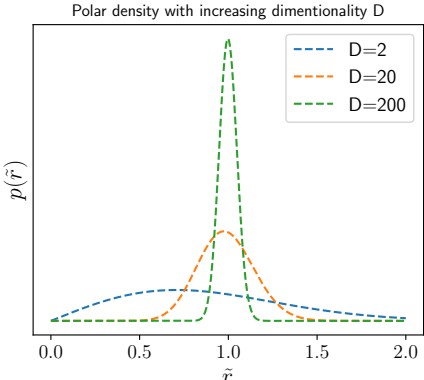 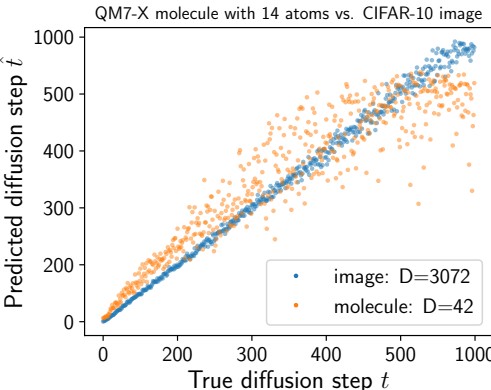

Figure A1: **Left:** (Scaled) Radius distribution (6) of Gaussain in the polar coordinate, illustrating the sharper density with increasing dimensionality $D$. **Right:** Diffusion time prediction performance on a molecule from QM7-X (orange) and an image from CIFAR-10 (blue).

which gives Eq.(6).

**Discussion:**

Our diffusion time step predictor essentially predicts the noise level of the current sample. Namely, it has to predict the variance of the Gaussian noise from only a SINGLE perturbed sample. Our intuition in a low-dimensional space would warn us that it is too challenging. However, our intuition does not really apply to high-dimensional spaces.

As discussed in Section 3.2, the marginal distribution (6) of the radius in the polar coordinate (see Figure A1 left) implies that most of the Gaussian perturbed samples lie in a thin shell with an equal distance to the Gaussian center. This implies that a neural network can learn to predict perturbation variance, and hence the diffusion time step, if it is capable of learning the data manifold, and hence can estimate the distance from a sample to the data manifold.

One might still worry that overlapping two Gaussian components for two training samples can disturb the diffusion time prediction. Indeed, the overlapping makes it hard to predict samples far from the training points, which can be empirically seen in Figure A1 right. However, overlapping does not significantly affect the prediction performance for small noise (hence small diffusion time step) samples in the neighborhood of training samples, again due to the high dimensionality. Assume that there are two training molecules $\boldsymbol{x}_a, \boldsymbol{x}_b$ with the distance $r = \|\boldsymbol{x}_a - \boldsymbol{x}_b\|$, and consider Gaussian blurred samples of $\boldsymbol{x}_a$ with the standard deviation equal to $\sigma = r$. Although, in this situation, $\boldsymbol{x}_b$ lies in the high-density shell of the Gaussian blur of $\boldsymbol{x}_a$ (the bump in Figure A1 left), the Gaussian blurred samples of $\boldsymbol{x}_a$ are uniformly distributed in the still high $(D - 1)$-dimensional shell, and therefore the probability that the Gaussian blur produces a sample close to $\boldsymbol{x}_b$ is extremely low. On the other hand, many training samples from the neighborhood of $\boldsymbol{x}_b$ are fed to the diffusion time predictor as low noise samples from $\boldsymbol{x}_b$. Accordingly, the network is trained so that they recognize the molecules close to $\boldsymbol{x}_b$ as low noise samples without being disturbed by high noise samples from $\boldsymbol{x}_a$. This intuition can be mathematically confirmed by computing the density ratio between the two mixture components around $\boldsymbol{x}_b$, i.e., $\mathcal{N}(\boldsymbol{x}_b + \boldsymbol{\varepsilon}; \boldsymbol{x}_b, \delta^2)/\mathcal{N}(\boldsymbol{x}_b + \boldsymbol{\varepsilon}; \boldsymbol{x}_a, r^2)$ for $\|\boldsymbol{\varepsilon}\| \sim \delta \ll r$, which is extremely high unless $D$ is very small.

This way, the high dimensionality enables accurate diffusion time step prediction especially for the samples close to stable molecules.

### A.2.2 EQUIVARIANT DIFFUSION FOR MOLECULAR STRUCTURES

Figure A2 gives a comprehensive illustration of the equivariant diffusion model used in this work based on the DDPM notation explained in Appendix A.1.1

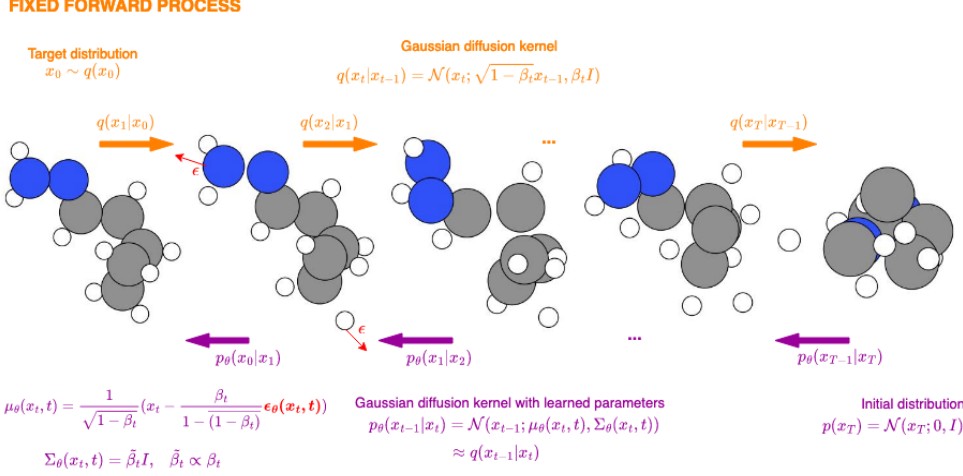

Figure A2: Scheme illustrating the diffusion model for molecular structures. It incorporates two stochastic processes, a forward and a backward process. The forward process involves a fixed Gaussian diffusion process with a Markovian kernel, $q(\mathbf{x}_t|\mathbf{x}_{t-1})$. This process transforms the original sample into Gaussian noise, effectively mapping a complex distribution $q(\mathbf{x}_0)$ to a simpler one $p(\mathbf{x}_T)$. On the other hand, the backward process aims to learn to reverse the forward process, i.e. $p_\theta(\mathbf{x}_{t-1}|\mathbf{x}_t) \approx q(\mathbf{x}_{t-1}|\mathbf{x}_t)$. Hence, it learns to map a simple prior distribution to a target distribution, allowing for the generation of novel molecules from noise. Refer to Appendix A.1.1 for more details about both processes and the different formulas.

### A.2.3 ALGORITHMS

Algorithm 1 shows the training procedure for MoreRed-JT. For the other two variants, we instead train two separate architectures and use only the first part of the loss in line 7 to train the denoising model and the second part to train the time step prediction model. Algorithm 2 describes the sampling with the adaptive MoreRed variants (AS and JT). MoreRed-ITP uses a fixed schedule but starts from a predicted time step instead of a fixed value.

---

**Algorithm 1** Training

**Input:** $q(\mathbf{x}_0)$, $\theta$, $a(t)$, $\alpha$, $\beta$
**Output:** $\varepsilon_\theta$, $g_\theta$
1: **repeat**
2: $\quad \mathbf{x}_0 \sim q(\mathbf{x}_0)$
3: $\quad t \sim \mathcal{U}(1, T)$
4: $\quad \varepsilon \sim \mathcal{N}(\mathbf{0}, \boldsymbol{I})$
5: $\quad$ subtract CoG for $\varepsilon$ if molecules
6: $\quad \mathbf{x}_t = \sqrt{\bar{\alpha}_t}\mathbf{x}_0 + \sqrt{1-\bar{\alpha}_t}\varepsilon$
7: $\quad$ Take gradient descent step on
$\quad\quad \nabla_\theta \|\varepsilon - \varepsilon_\theta(\mathbf{x}_t, g_\theta(\mathbf{x}_t))\|^2$
$\quad\quad + (1-\eta)\|g_\theta(\mathbf{x}_t) - a(t)\|^2$
8: **until** convergence

---

**Algorithm 2** Sampling

**Input:** $\varepsilon_\theta$, $g_\theta$
**Output:** new sample $\mathbf{x}_i$, #iterations $i$
1: $i = 0$
2: $\mathbf{x}_i \sim \mathcal{N}(\mathcal{N}(\mathbf{0}, \boldsymbol{I}))$
3: **while** $g_\theta(\mathbf{x}_i) \neq 0$ **do**
4: $\quad \hat{t} = g_\theta(\mathbf{x}_i)$
5: $\quad \mathbf{z} \sim \mathcal{N}(\mathbf{0}, \boldsymbol{I})$ if $\hat{t} > 1$, else $\mathbf{z} = 0$
6: $\quad$ subtract CoG for $\mathbf{z}$ if molecules
7: $\quad$ subtract CoG for $\varepsilon_\theta$ if molecules
8: $\quad x_{i+1} = \frac{1}{\sqrt{1-\beta_{\hat{t}}}}(\mathbf{x}_i + \frac{\beta_{\hat{t}}}{\sqrt{1-\bar{\alpha}_{\hat{t}}}}\varepsilon_\theta(\mathbf{x}_i, \hat{t})) + \sqrt{\beta_{\hat{t}}}\mathbf{z}$
9: $\quad i = i + 1$
10: **end while**
11: **return** $\mathbf{x}_i$, $i$

---

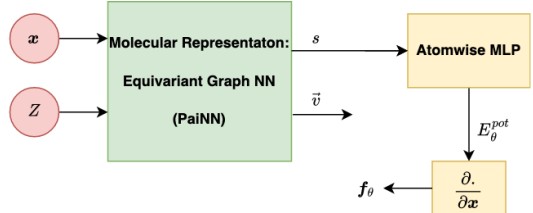

Figure A3: The force field model architecture: $\mathbf{x}$ are the atom positions, and $\mathbf{Z}$ are the atom types. It uses PaiNN as molecular representation. The scalar features $s$ are used to estimate the energy $E_\theta^{pot} = \sum_{i=1}^{N} E_\theta^{(i)}$, which is derived with respect to the positions $\mathbf{x}$ to compute the forces $\boldsymbol{f}_\theta = -\frac{\partial E_\theta^{pot}}{\partial \mathbf{x}}$.

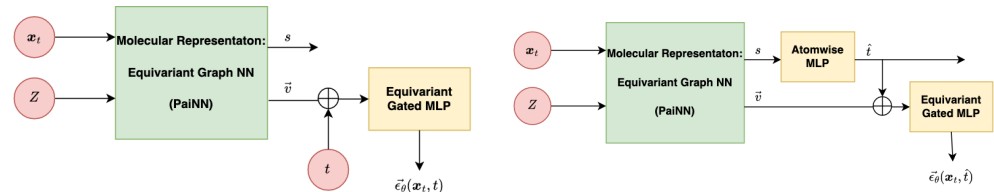

Figure A4: $\mathbf{x}_t$ are the atom positions at diffusion step $t$, and $\mathbf{Z}$ are the atom types. Both models use PaiNN as an equivariant molecular representation. **Left:** The noise model architecture used for DDPM, MoreRed-AS and MoreRed-ITP. The diffusion step $t$ is normalized by $t/T$ and concatenated to the vectorial features $\boldsymbol{v}$ to directly predict the noise $\boldsymbol{\varepsilon}_\theta(\mathbf{x}_t, t)$ using an equivariant gated MLP. **Right:** The model architecture for MoreRed-JT. It is the same as the other architecture except that the time step is first predicted by an atomwise MLP on top of the invariant scalar features $\boldsymbol{s}$, similar to predicting the invariant potential energy in force field models.

## A.3 MODEL ARCHITECTURES AND HYPERPARAMETERS

### A.3.1 ARCHITECTURE

**Force field model:** The architecture of the force field model is illustrated in Figure A3. Based on the invariant feature representation, a multi layer perceptron (MLP) output head predicts the atomwise energies $\{E_\theta^{(i)}\}_{i=1}^{N}$, which are aggregated to form a permutation-invariant potential energy $E_\theta^{pot} = \sum_{i=1}^{N} E_\theta^{(i)}$. The gradients, i.e., the interatomic forces, are computed as a derivative of the molecular energy with respect to the atom positions, which ensures equivariant predictions of interatomic forces.

**MoreRed variants, including DDPM:** The architectures of the different variants are illustrated in Figure A4. For MoreRed-JT, we use the architecture on the right side of Figure A4 where the time and noise head share the same representation. For DDPM, we use the architecture on the left side of Figure A4, which predicts the noise conditioned on the time step $t$. For MoreRed-ITP and MoreRed-AS, we use the same noise model architecture as for DDPM, with the only difference being that the time $t$ is being predicted by a separate network via an atomwise MLP head, similar to the time predictor in MoreRed-JT but without a joint noise model.

### A.3.2 HYPERPARAMETERS

The models used in our experiments with molecules were implemented and trained using SchNet-Pack (Schütt et al., 2023). For the molecular representation, we used PaiNN (Schütt et al., 2021) with 3 interaction blocks and 20 Gaussian radial basis functions with a cutoff of 5Å for all the models. After computing the molecular representation, the number of atomic features is halved in each layer of the output heads, with a total of 3 layers for each head for all the models. We use the AdamW optimizer for all the models and we train them until complete convergence. For force

field training, we follow the hyperparameters and the training details reported in the original work (Schütt et al., 2021).

**Force field model:** We tuned the batch size on $\{10, 64, 128\}$, the learning rate on $\{10^{-3}, 10^{-4}\}$ and the number of atomic features on $\{64, 128, 256\}$. We found that a batch size of 10, learning rate of $10^{-3}$ and 128 atomic features achieve the lowest loss, which aligns with the results from previous work using PaiNN (Schütt et al., 2021; Unke et al., 2021a). We found that using more atomic features than 128, i.e. more parameters, for the force field model hurt the performance. We believe that increasing the number of parameters while having a fixed number of data does not increase the results anymore and increases the risk of fast overfitting.

For the training of the force field model, we used the exponential moving average (EMA) of the parameters across all training epochs during validation and testing rather than using the most recent parameter updates. Additionally, the learning rate was halved during training if the validation loss stagnated for 15 epochs, allowing for finer steps near local minima and avoiding fluctuation around them. Besides, we used early stopping to stop the training process when the validation loss stopped decreasing for 30 epochs instead of using a fixed number of epochs.

**MoreRed variants, including DDPM:** We used a large batch size of 128 to improve the accuracy of the loss, as it involved uniformly sampling a single diffusion step per molecule per batch because we sample one diffusion step per molecule instead of the whole trajectory. We set the number of atomic features to 256 for all variants of MoreRed, except MoreRed-large and DDPM-large, where we use 512 atomic features and 5 layers for the noise head instead of 3, resulting in circa $10M$ parameters instead of $2.5M$. For MoreRed-JT, we found that setting $\eta$ to $0.9$ works well because the noise prediction provides more signals, $3N$ per molecule with $N$ atoms, compared to diffusion step prediction, one step per molecule. While we used a separate time predictor with a separate representation for MoreRed-AS and MoreRed-ITP, all hyperparameters were kept consistent across all the models for all experiments and datasets, and we trained all of them until complete convergence.

During the training of MoreRed models, we sample one time step $t$ for each molecule, with the total number of molecules used per training iteration equal to the batch size, i.e. for each training epoch, we use one uniformly sampled diffusion step per molecule. As depicted in Figure A5, this approach resulted in a noisy training loss since the model used different diffusion steps at each training iteration instead of the entire diffusion trajectory. To mitigate this issue, we used EMA of the parameters across all training epochs during validation and testing rather than using the most recent parameter updates. This approach yielded smoother learning evolution, as reflected in the validation loss because the EMA of the parameters better maintains the previously learned signal from the different diffusion steps seen per molecule.

Additionally, similar to force field model training, the learning rate was halved during training if the validation loss stagnated for 150 epochs instead of 15 because MoreRed uses only unstable molecules resulting in 100 times fewer data and fewer iterations per epoch. Moreover, we used early stopping with 300 epochs.

In initial experiments with MoreRed, we observed that the model outputs occasionally explode, resulting in exploding gradients and divergence of the training, as illustrated in Figure A5 after 400 epochs. Subsequent analysis revealed that this phenomenon occurs only for high diffusion steps above $t = 850$, which was associated with highly noisy and unrealistically dense molecules where all atoms were tightly packed with small atomic distances, as shown in Figure A5. Hence, we added gradient clipping with a value of $0.5$ to mitigate this explosive behaviour.

**Details for SO3Net:** We use the same hyperparameters for the force field model and all variants of MoreRed. Specifically, we maintain a consistent configuration with 128 atomic features across the entire architecture and utilize 2 hidden layers for all property prediction heads. Additionally, we set $l_{\max} = 1$ for the maximum degree of the spherical harmonics features. To speed up training, a large batch size of 512 and a learning rate of $2 \cdot 10^{-3}$ is utilized for all models. All other experimental details remain unchanged from those employed in the PaiNN experiments.

**UNet for image generation:** For the image generation task, we employ the UNet architecture for the noise model, where we follow the hyperparameters and training choices of Ho et al. (2020) for the linear noise schedule and of Nichol & Dhariwal (2021) for the cosine noise schedule. We use the UNet model implementation from the Hugging Face framework (von Platen et al., 2022). For

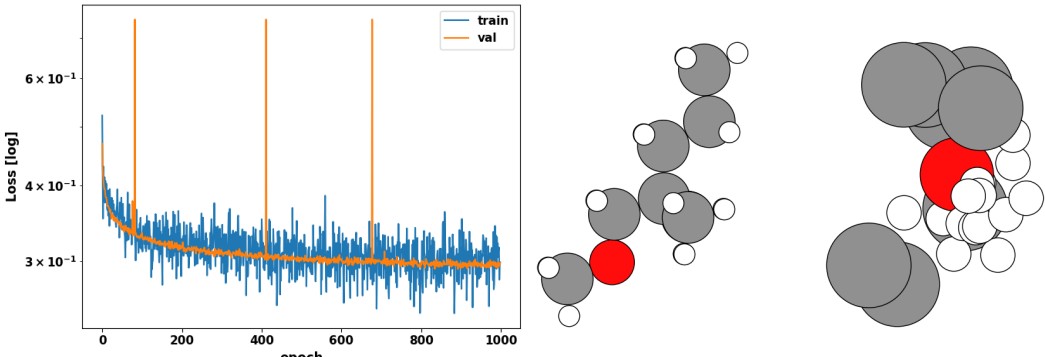

Figure A5: **Left:** Plot showing the training loss in blue and validation loss in orange in the log scale. The training loss has more fluctuations than the validation noise. The validation loss shows three high peaks due to exploding model outputs. **Right:** Example of a degenerated molecule after 916 diffusion steps. On the left is the original molecule. On the right is the diffused molecule, where all the hydrogens (white nodes) are tightly condensed.

the diffusion time step predictor, we duplicate the same UNet as for the noise model and predict one time step per pixel. These pixel-level predictions are aggregated, and the average value is used as one time step estimate per image.

## A.4 EXPERIMENTS: DETAILS

### A.4.1 DATASETS

**QM7-X:** QM7-X (Hoja et al., 2021) is a comprehensive dataset that was derived from 7k molecular graphs sampled from the GDB13 chemical space with up to 7 heavy atoms, including types C, N, O, S, and Cl. For each SMILE string structural and constitutional isomers were obtained using the MMFF94 force field and subsequently optimized with accurate DFT simulations, leading to 42k stable structures. To capture the potential energy surface close to the stable molecules, unstable molecules were generated by displacing each stable molecule along a linear combination of normal mode coordinates computed at the DFTB3+MBD level, such that the energy difference between the unstable and stable structures follow a Boltzmann distribution. For each stable structure, 100 unstable configurations were generated, leading to 4.2M unstable structures in total.

For our experiments, data splitting for training, validation, and testing is done at the molecular graph level to prevent bias leakage between different splits due to related isomers and conformations originating from the same graph. Specifically, we use the molecules resulting from 4500 graphs for training, 1250 for validation and the rest for testing. Note that MoreRed does not utilize the unstable configurations for training, effectively decreasing the training set size by a factor of 100 compared to training set of the FF model.

Although the dataset might differ from a real-world distribution, where the candidates are obtained with cheap optimization methods, it currently is the only dataset that provides a large number of stable molecules with various compositions, as well as corresponding unstable variants of these with labels for energy and forces, which are necessary for a fair comparison of our method to machine learning force fields on the task of molecule optimization. Besides, both QM7-X and cheap optimization methods in real-world scenarios give physically feasible unstable structures that cover the potential energy surface with non-Gaussian noise, and therefore we expect that our results are still indicative for the realistic scenario.

### A.4.2 EXTENDED ANALYSIS FOR THE TIME STEP PREDICTION

Figure A1 (right) with orange dots shows a scatter plot of the true diffusion time vs. its prediction by our diffusion time predictor after training on the stable molecules from the QM7-X dataset. We also show performance of the diffusion time predictor trained on CIFAR10, where the images have

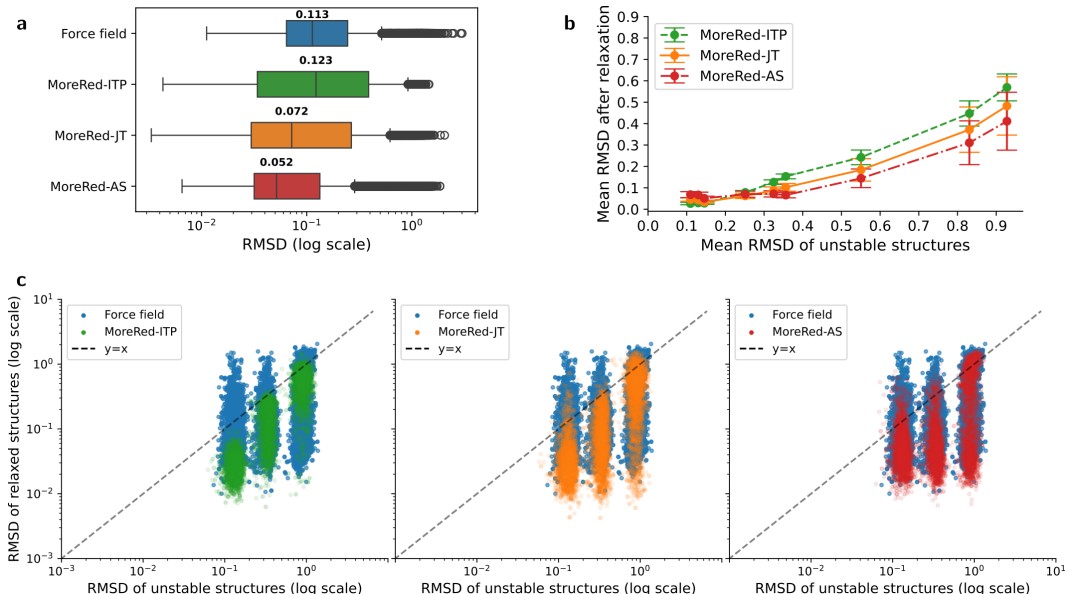

Figure A6: **a**: The RMSD of molecules relaxed with the baseline FF model and with our MoreRed variants for 20k unstable structures from the QM7-X test split. **b**: The mean RMSD of 20k unstable structures from the QM7-X test split vs. their mean RMSD after relaxation. The mean is taken over bins of 2k unstable molecules with increasing RMSD. The bars show the standard deviation of the RMSD after relaxation over five runs with the respective MoreRed variant. **c**: RMSD of 10k unstable structures from the QM7-X test split before (x-axis) and after (y-axis) relaxation for all three variants of MoreRed and the baseline FF model. The three partitions occur because the test structures are sampled to cover low/intermediate/high RMSD values (see the **Results** paragraph in Section 4.2 for details on the sampling procedure).

a higher dimensionality than the molecule data, as blue dots.[2] As discussed in Section 3.2, the diffusion time prediction is easier when the dimension $D$ is large, and the true diffusion time, i.e., the noise level, is smaller.

### A.4.3 EXTENDED ANALYSIS OF RELAXATION WITH MORERED

Here we further analyze the three different variants of MoreRed by discussing extended results from our experiments on relaxing unstable structures from the QM7-X test set described in Section 4.2. In Figure A6, we analyze the RMSD values of the optimized molecules in comparison to their stable structure, instead of the RMSD ratio as it is reported in the main text. First of all, in Figure A6a we compare the RMSD values of the three MoreRed variants to the force field. Notably, the two variants with adaptive scheduling outperform the baseline FF as in Figure 3a, while the median RMSD of MoreRed-ITP is slightly worse than that of the baseline FF. The reason for this can be seen in Figure A6c, where the RMSD values after relaxation are compared to the RMSD values of the initial unstable structures for all three MoreRed variants. While the performance of MoreRed-ITP (green) is particularly good for structures that are already close to the stable state, its performance is impaired for structures that initially have a high RMSD. The adaptive variants, MoreRed-JT/-AS (orange, red), show a more balanced performance, successfully relaxing structures over the whole spectrum of unstable test molecules. This suggests that the adaptive scheduling with the time step prediction improves relaxation of molecules that are further away from the data manifold, which is in line with our findings in Section 4.1. This comes at the cost of a higher number of relaxation steps for the adaptive variants and more failure cases (see Section 4.2 and Figure 3).

Furthermore, to investigate the stochasticity of our method, we analyze the mean RMSD values and their standard deviation from the mean after optimization, subject to the RMSD of the initial structures. For this we created 8 bins of initial structures based on their RMDS and measured the

---

[2]QM7-X and CIFAR10 datasets are explained in Sections 4.2 and 4.3, respectively.

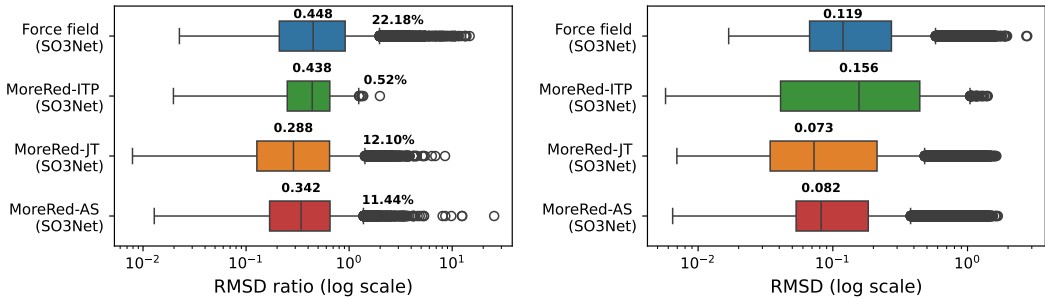

Figure A7: The RMSD ratios (left) and RMSD (right) of molecules relaxed with the baseline FF model and with our MoreRed variants for 20k unstable structures from the QM7-X test split using SO3Net implementation in SchNetPack (Schütt et al., 2023) as equivariant backbone representation. The median values and the percentage of relaxation failure cases, i.e., the cases where the RMSD ratio exceeds 1, are shown above each box plot.

RMSD after optimization (see Figure A6b). It shows that not only the mean RMSD increases based on the initial RMSD, but also the standard deviation of the RMSD after optimization increases. This is expected, because structures with large RMSD are assigned to high time steps, resulting in higher variance values for the diffusion reverse kernel. Besides, with high time steps, MoreRed needs more optimization steps until convergence and with every step a small amount of stochasticity is added to the positions. Considering statistics over all test structures, the deviations across multiple relaxation runs with MoreRed are very low and therefore not reported in the boxplots.

As a conclusion, for initial structures with higher levels of perturbation, the RMSD of the optimized structures increases and the variants with adaptive scheduling, MoreRed-JT/-AS, via time step prediction provide the required flexibility to perform accurate optimization. On the other hand, the fixed schedule variant, MoreRed-ITP, provides fast and very accurate stable molecules for initial structures with low levels of noise.

### A.4.4 GENERALIZATION WITH DIFFERENT EQUIVARIANT REPRESENTATIONS

To further assess the robustness of our method, we conducted a set of experiments employing an alternative equivariant molecular representation. Specifically, we trained a force field model and all three variants of MoreRed using SO3Net (Schütt et al., 2023) as a backbone representation. This representation incorporates spherical harmonics in the spirit of Tensor Field Networks (Thomas et al., 2018) and NequIP (Batzner et al., 2022) to handle SO(3)-equivariance, distinguishing it from the PaiNN architecture.

Utilizing the same data splits as in the PaiNN experiments, we tested the models on the same set of 20k unstable structures from the test split of QM7-X. All other experimental details align with those outlined in section 4.2 for PaiNN. Given the long training time of the force field model (7 days), we opted to use half the number of parameters employed in PaiNN to expedite the experiments. However, to maintain fairness, we used the identical model hyperparameters for both MoreRed and the force field. Additional hyperparameter details are provided in section A.3.2.

Our findings, summarized in Figure A7, affirm that our approach performs comparably well with this alternative equivariant neural network backbone, consistently outperforming the force field model in terms of structure accuracy with MoreRed-AS and MoreRed-JT. Yet, the overall performance for all models, including the force field, is slightly worse than reported in Figure 3 with PaiNN, and there is a subtle discrepancy in performance between MoreRed-AS and MoreRed-JT. We attribute these differences to the lack of hyperparameter tuning with SO3Net and the use of half the number of parameters employed in PaiNN.

### A.4.5 DATA GENERATION WITH TIME STEP PREDICTION

**Improved Unconditional Generation with Adaptive Schedule:** In the relaxation experiment in Figure 3 , More-Red-AS/-JT, utilizing adaptive scheduling by predicting the diffusion time steps

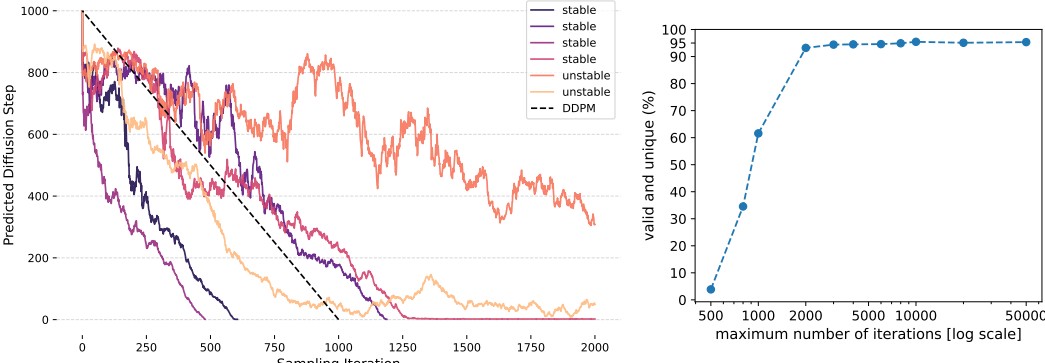

Figure A8: **Left:** Examples of sampling trajectories of new stable/unstable molecular structures from complete noise using MoreRed-JT. The solid lines indicate the predicted time steps through the sampling iterations. The dashed line indicates the fixed decreasing time step schedule used in the usual reverse diffusion, e.g. DDPM. Each color indicates a different sample. Right: The evolution of validity and uniqueness when increasing the maximum number of iterations for MoreRed-large in the molecule generation task on QM9.

throughout the entire sampling trajectory, outperforms MoreRed-ITP, which predicts only the starting step and employs a fixed decreasing schedule akin to plain reverse diffusion. This observation extends to new sample generation from complete noise, as supported by the results in Table 1 and 2. They demonstrate that adding the time step predictor to control the sampling trajectory while maintaining the same noise head from DDPM models significantly enhances the results of molecular structure and image generation.

Moreover, similar to the trend seen in the relaxation trajectories in Figure 2b, the sampling trajectories in Figure A8 (left), which start from complete noise, further underscore the flexibility introduced by the time step prediction. It predicts the current noise level, which correlates with the distance to the target data manifold, to adjust subsequent denoising time steps. This adaptation compensates for errors introduced by the noise predictor and accumulated through previous denoising steps. If needed, the process can move back in time and take as many steps as required to converge to the data manifold. Moreover, the time step prediction may jump forward in time if the current sample is closer to the data manifold than expected and thus a lower diffusion time step is predicted.

As a motivation for this idea, previous work (Song et al., 2021) has identified issues with fixed reverse diffusion models, where a potential error in sample $x_t$ at time step $t$, due to using one noise prediction per time step, can deviate the sample from the optimal reverse trajectory. If uncorrected, this error may propagate, get amplified and result in worse sample quality. Song et al. (2021); Karras et al. (2022) partially mitigate this by incorporating correction steps after each diffusion reverse step, such as using second-order SDE/ODE solvers or running Langevin dynamics iterations after each reverse step $t$. While these approaches can significantly improve the sample quality and using more MCMC Langevin iterations during sampling can improve the results, they require tuning additional hyperparameters, including the number of correction steps per reverse step, Langevin damping rate, and total denoising iterations. Besides, Karras et al. (2022) demonstrated that too many correction steps can harm sample quality and that exhaustive hyperparameter tuning may be required for each model and dataset, as reported in their experiments using different datasets and models. We believe that this extends to even each sampling trajectory. Hence, our adaptive scheduling strategy offers a promising solution, dynamically determining the number of reverse iterations during each sampling trajectory and automatically adjusting the correction steps based on the mismatch between current and predicted diffusion time and the implicit distance from the data manifold. Table 1 and 2 and Figure 2 and A8 further support this hypothesis, showcasing significant improvements in molecule relaxation and unconditional molecular structure and image generation.

**Maximum Number of Sampling Iterations:** The results presented in table 1 are obtained using a maximum number of sampling steps of 2000. However, to gain a more comprehensive understand-

Table A1: An extended comparison of our method to previous work on molecule generation using the QM9 dataset. *Unlike MoreRed, previous work generate not only the atom positions but also the atomic compositions.

| Model | Valid (%) | +Unique (%) | +Novel (%) |
|---|---|---|---|
| DDPM | 78.2 | 77.3 | 62.5 |
| MoreRed | **89.3** | **88.0** | **68.6** |
| DDPM-large | 86.6 | 85.3 | 63.8 |
| MoreRed-large | **94.7** | **92.4** | **66.7** |
| GeoLDM* (Xu et al., 2023) | **93.8** | **92.7** | - |
| EDM* (Hoogeboom et al., 2022) | 91.9 | 90.7 | 65.7 |
| G-SchNet* (Gebauer et al., 2019) | 85.5 | 80.3 | 63.8 |
| E-NF* (Garcia Satorras et al., 2021) | 40.2 | 39.4 | - |

ing of the model's behaviour, we conduct experiments with varying maximum sampling steps, and the validity and uniqueness results are summarized in Figure A8.

We observe that the model can generate around $4\%$ of valid and unique molecules with less or equal to $500$ sampling iterations and up to $35\%$ with no more than 800 steps. Interestingly, MoreRed-large yields around $25\%$ fewer valid and unique molecules compared to the standard diffusion model DDPM-large when using $1000$ steps. Yet, it outperforms it by approximately $7\%$ using 2000 sampling steps at most. This suggests that the error introduced by the stochastic predictor of the diffusion step may slow down convergence in certain cases but ultimately leads to finer samples. Furthermore, by observing the evolution of the curve, we deduce that using 2000 steps is sufficient to achieve results close to the best performance. Beyond 2000 steps and up to 50k sampling steps, it exhibits only marginal improvements compared to the significant progress observed between $500$ and $2000$ steps. The latter aligns with the findings of Song et al. (2021), who showed that using 2000 steps of a predictor-corrector sampler instead of only 1000 steps predictor-only sampler to sample from a diffusion model trained on 1000 steps enhances performance in images. Yet, in our method, we do not fix the number of iterations to 2000 for all the samples but MoreRed dynamically sets the number of steps by predicting the time step and ending sampling when convergence criteria, $\hat{t} \leq 0$, is met.

**Comparison to other Molecule Generation Methods on QM9:** As our primary focus is on structure relaxation, we model the conditional probability $p(\mathbf{x} \mid \mathbf{Z})$, which allows generating molecular structures $\mathbf{x}$ in 3D space given a valid atomic composition $\mathbf{Z}$. A direct comparison with previous work for molecule generation on QM9 is not possible because, unlike our approach, previous approaches model the joint distribution $p(\mathbf{x}, \mathbf{Z})$ to simultaneously generate both atom types and positions. Yet, for completeness, we report the results from previous work in Table A1 as an extension of Table 1.

### A.4.6  COMPUTATION TIME

Here we give a more detailed analysis of the computation time for the three MoreRed variants and the force field model that have been used for the experiments in section 4.2. As mentioned in the main text, the median number of steps until convergence as well as the accuracy of the stable structures strongly depend on the MoreRed variant, ranging from 53 steps for the fastest MoreRed-ITP until 1000 steps for the most accurate MoreRed-AS. For MoreRed-AS, we observe many trajectories where the model predicts many consecutive low time steps until reaching the maximum number of allowed steps, but the optimization does not converge due to the strict convergence criterion of $t = 0$. Applying less strict convergence criteria might decrease the number of steps per structure optimization significantly, which is a direction for future work. Furthermore the computation time per structure during inference dramatically improves if many structures are evaluated in batches instead of sequentially. While batch-wise optimization can be done straight forward with all MoreRed variants, batch-wise structure optimization utilizing the force field model is not trivial, due to the dependency on the L-BFGS optimizer. In the following we give a rough estimate of the inference time by either using sequential optimization or batch-wise optimization. For the analysis, we neglect any computational cost that is not directly related to model inference.

**Sequential Optimization** Comparing MoreRed variants to the force field model, we observed an average inference time for a single structure, not a batch, of 0.03s for MoreRed and 0.02s for the

force field model. To compute the mean inference time per structure optimization performed sequentially, we need to use not the median, as reported in Section 4.2, but the mean. The mean number of optimization steps until convergence was measured as 64 steps for MoreRed-ITP, 489 steps for MoreRed-JT, 992 for MoreRed-AS, and 122 for the force field model, which results in an average total inference time per structure optimization of $0.03s \cdot 64 = 1.92s$ for MoreRed-ITP, 14.67s for MoreRed-JT, 29.76s for MoreRed-As and 2.44s for the force field.

**Parallelized Optimization** For efficient optimization of a large number of structures, as it is usually the case in many applications, model inference is preferably done in batches. Assuming batch-wise relaxation with the force field model is possible, we observed an inference time for evaluating a batch of 128 molecules of 0.03 s for the force field and 0.05 s for the MoreRed variants. Since the batch-wise relaxation is done until all the structures in the batch have converged, in a worst case scenario, which is more likely to happen the larger the batch is, both methods need the maximum number of allowed steps, 1000. This would result in an average inference time per structure optimization of $\frac{0.03s \cdot 1000}{128} = 0.23s$ for the force field and $\frac{0.05s \cdot 1000}{128} = 0.39s$ for the MoreRed variants.

### A.4.7    EXAMPLES OF GENERATED SAMPLES

**QM9:** Figures A9 and A10 showcase random samples of molecular structures generated by MoreRed and DDPM trained on QM9.

**CIFAR-10:** Figures A11 and A12 showcase random samples generated by MoreRed and DDPM, both trained on CIFAR-10 using the linear noise schedule proposed by Ho et al. (2020) with $T = 1000$. Similarly, Figures A13 and A14 display samples generated by MoreRed and DDPM, trained on CIFAR-10, but this time employing the cosine noise schedule introduced by Nichol & Dhariwal (2021) with $T = 1000$. Both MoreRed models use a convergence criterion set at $\hat{t} \leq 0$ or a maximum of 2000 steps.

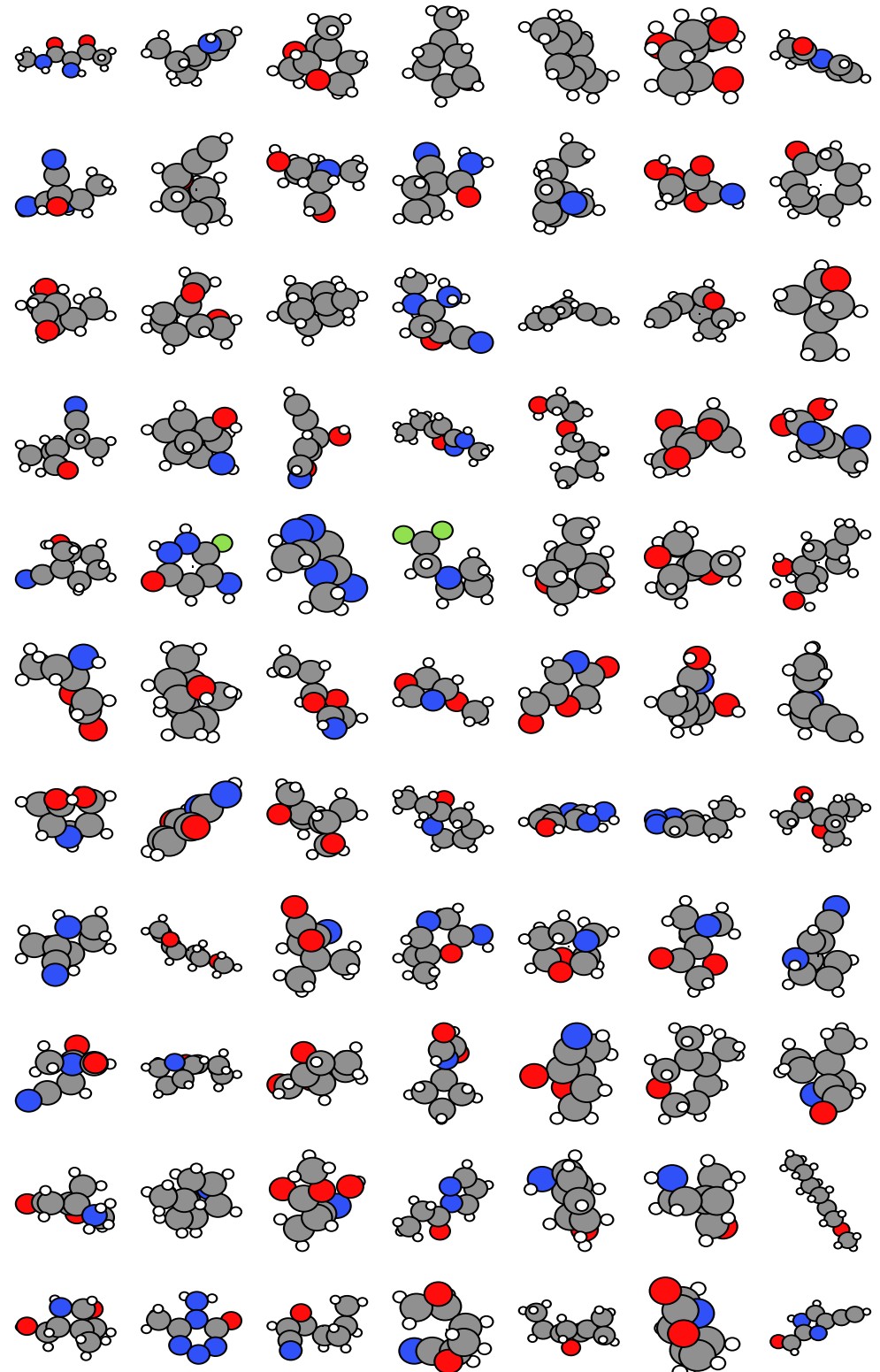

Figure A9: Batch of generated molecules with MoreRed trained on QM9

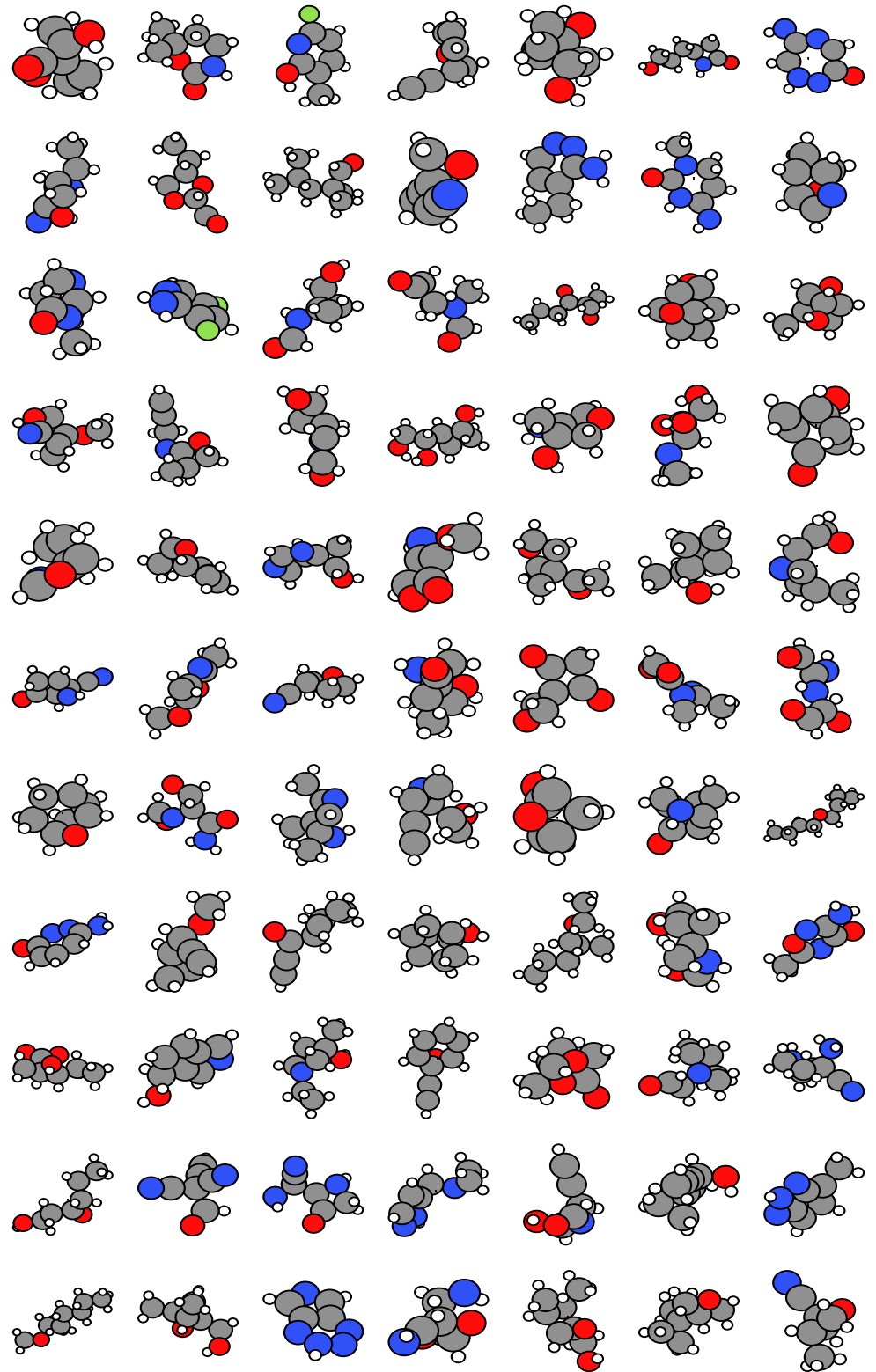

Figure A10: Batch of generated molecules with DDPM trained on QM9

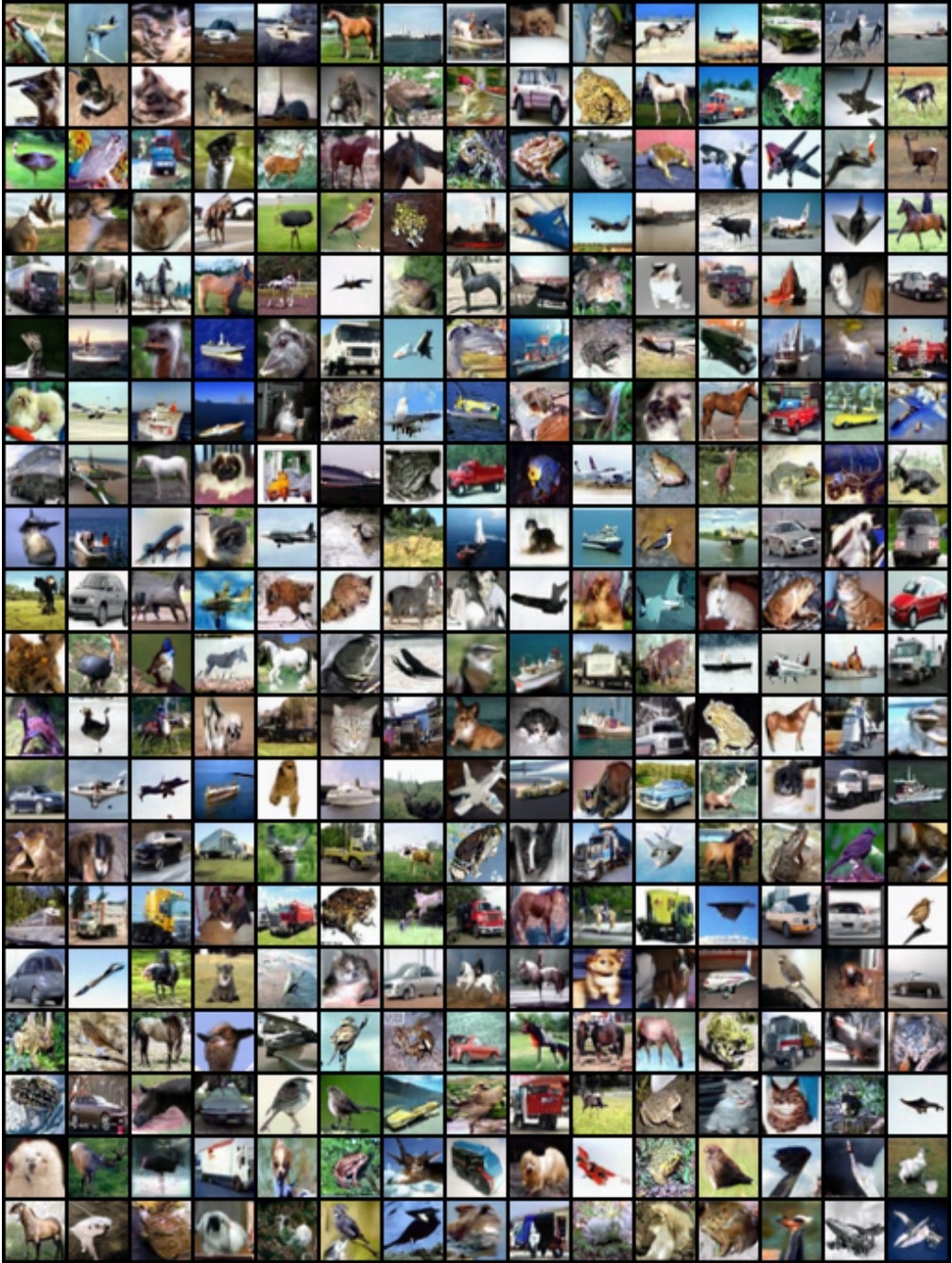

Figure A11: Batch of generated images with MoreRed trained on CIFAR-10 using linear noise schedule

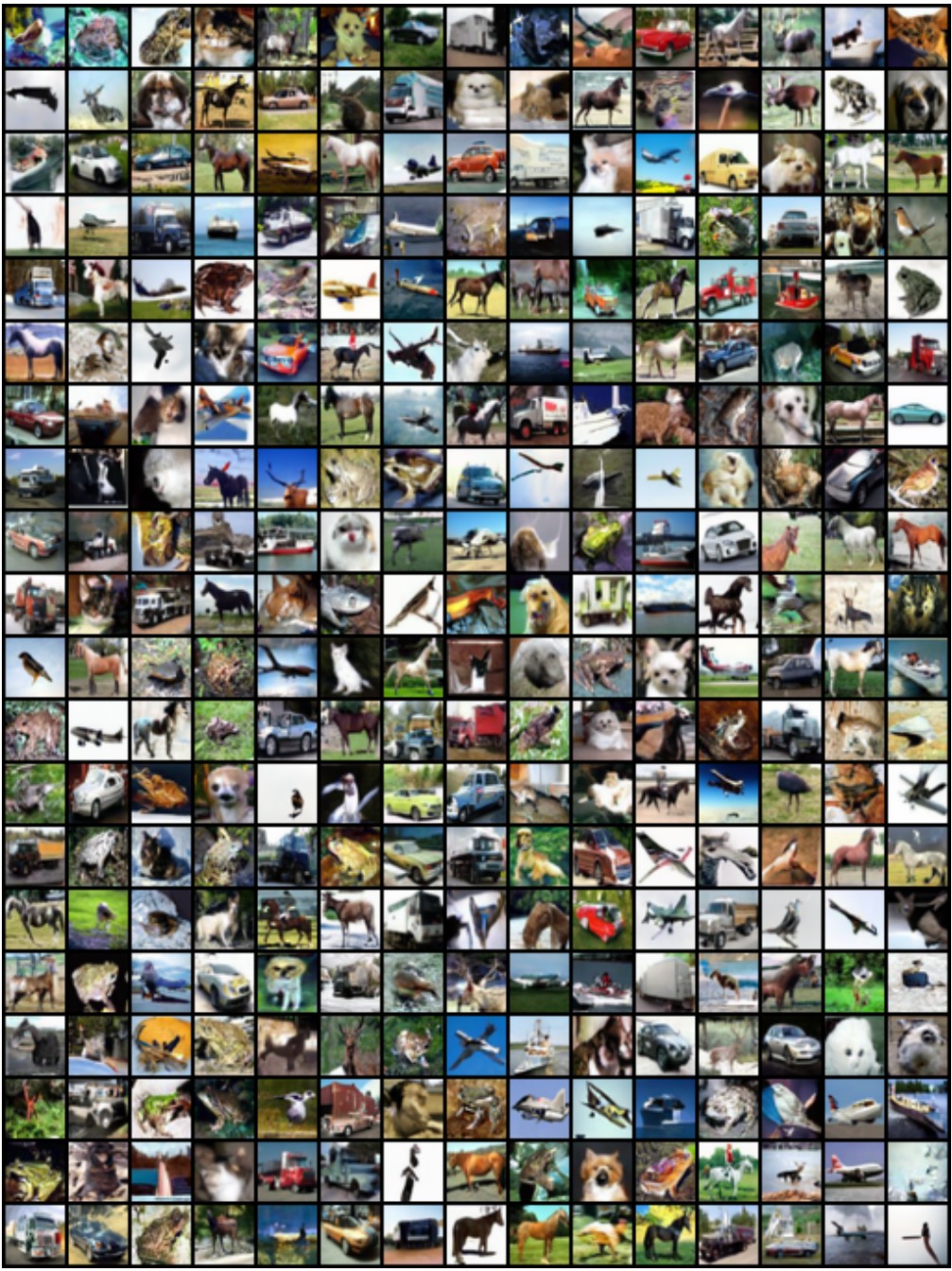

Figure A12: Batch of generated images with DDPM trained on CIFAR-10 using linear noise schedule

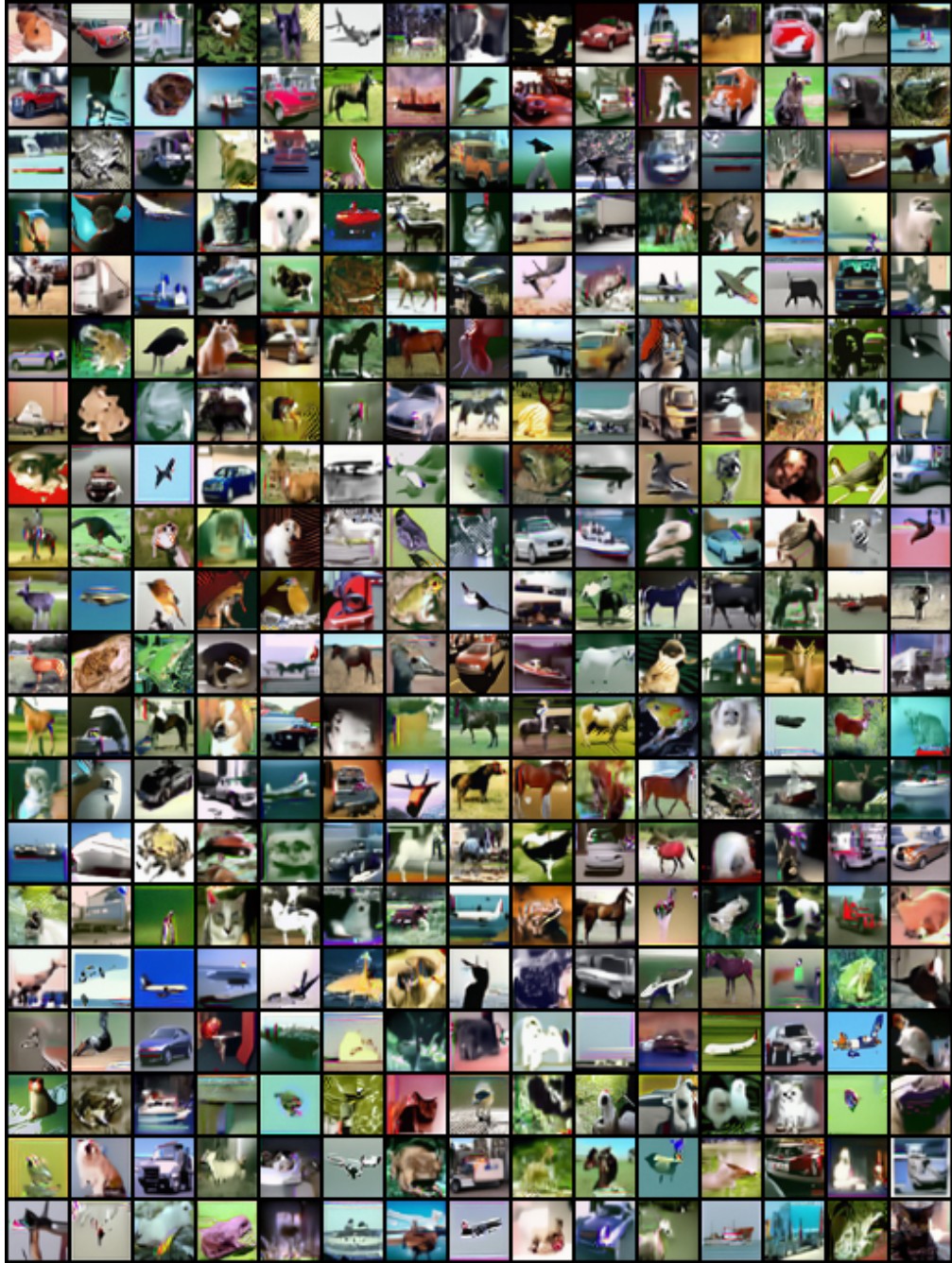

Figure A13: Batch of generated images with MoreRed trained on CIFAR-10 using cosine noise schedule

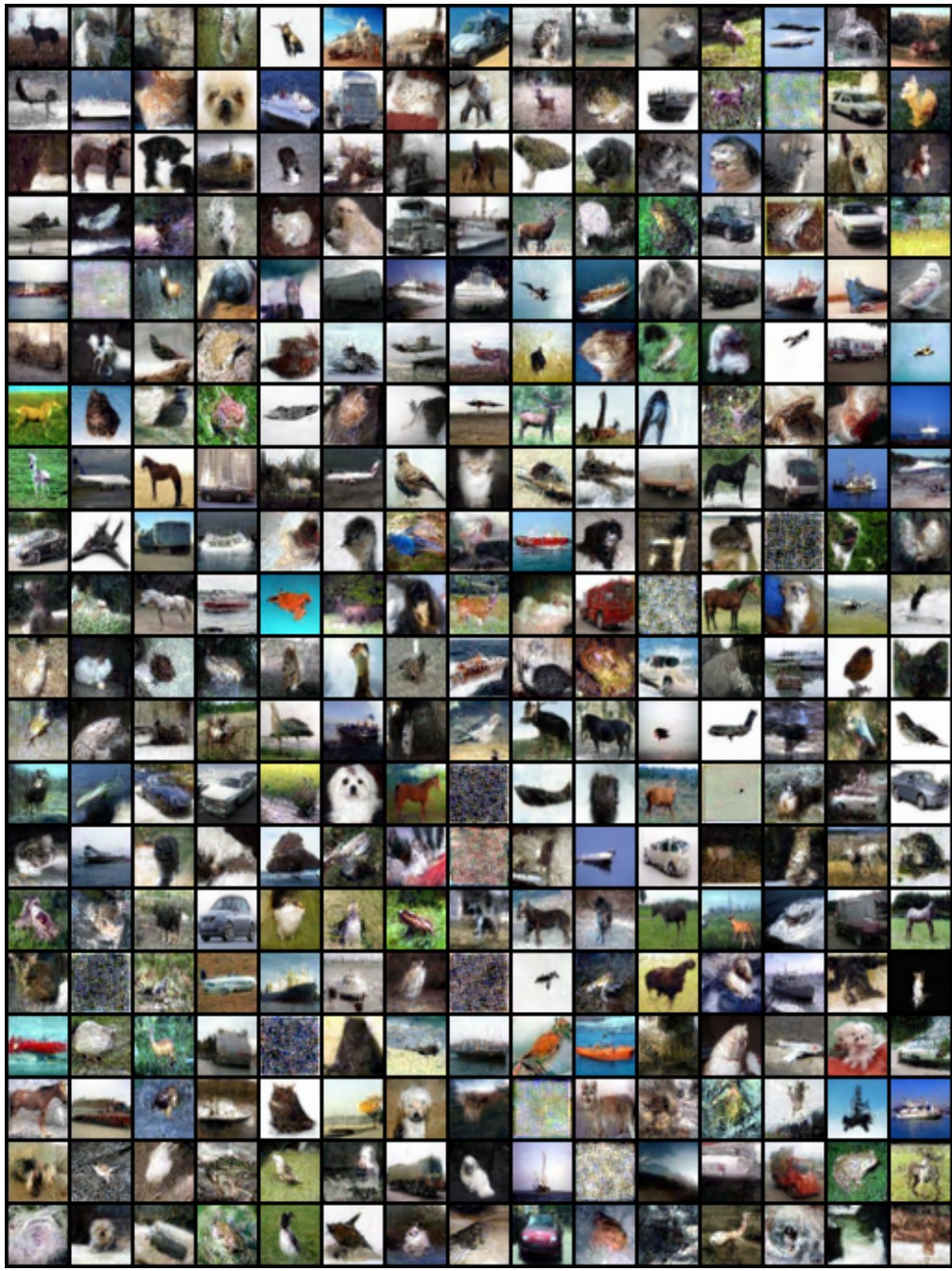

Figure A14: Batch of generated images with DDPM trained on CIFAR-10 using cosine noise schedule

