# OpenReview forum: "Molecule Relaxation by Reverse Diffusion with Time Step Prediction"
_ICLR.cc/2024/Conference — Submitted to ICLR 2024_

### Official Review · Reviewer_2WpS · 2023-10-17

**Soundness:** 3 good
**Presentation:** 3 good
**Contribution:** 2 fair
**Rating:** 6
**Confidence:** 5

**Summary:**

This paper focuses on the molecule relaxation task and proposes a method entitled MoreRed, where unstable molecules are seen as noisy samples to be denoised by a diffusion model equipped with a time step predictor to handle arbitrarily noisy inputs.

**Strengths:**

1. I appreciate the presentation of the paper; it is well-organized.
2. The research problem is highly relevant to the ICLR community.
3. The method is well-motivated.
4. The experimental results are significant.

**Weaknesses:**

1. I believe that the idea of using diffusion in this paper is quite straightforward and not very innovative, so I find this aspect lacking in novelty.
2. The proposed "Diffusion Time Step Prediction" seems to have a loose connection with the main task of molecule relaxation studied in the paper. The authors also mentioned that it appears to be a generic technique that could be applied in other domains like image generation.

If this paper solely relies on diffusion to address the molecule relaxation task, I think it lacks significant innovation. Furthermore, the "time step prediction" aspect doesn't seem closely related to the main task.

**Questions:**

The paper uses PaiNN as the backbone and compares it with force field (FF) methods also based on PaiNN. I would like to see the authors try using other different backbones to evaluate the robustness of their proposed method concerning the choice of backbone.

---

> ### Author Response · Authors · 2023-11-22
> **Answer 1/2 to Reviewer 2WpS**
>
> Dear Reviewer 2WpS,
>
> thank you for taking your valuable time and providing us with questions and feedback. In the following, we provide answers to your concerns and questions.
>
> > **Weaknesses:**
> >
> > 1. I believe that the idea of using diffusion in this paper is quite straightforward and not very innovative, so I find this aspect lacking in novelty.
> > 2. The proposed "Diffusion Time Step Prediction" seems to have a loose connection with the main task of molecule relaxation studied in the paper. The authors also mentioned that it appears to be a generic technique that could be applied in other domains like image generation.
> >
> > If this paper solely relies on diffusion to address the molecule relaxation task, I think it lacks significant innovation. Furthermore, the "time step prediction" aspect doesn't seem closely related to the main task.
>
> We respectfully disagree with the reviewer regarding a lack of novelty in our approach. The prediction of the diffusion time step is a novel contribution and applying diffusion as denoising to perform relaxation was not done before. Furthermore, the diffusion time predictor  is not “loosely connected” to the relaxation but a prerequisite for applying the diffusion model framework to relaxation: It enables the reverse-diffusion from arbitrarily noisy inputs, e.g. molecules that are close to their stable conformation. This is because diffusion models use the time step $t$ as an input, and therefore we must determine  from which $t$ the reverse diffusion should start for a given unstable molecule. We have added Fig. 2a to show how the predicted timestep relates to the RMSD of the unstable molecules (i.e. their distance to the ground-truth stable structure). One can see that the time step prediction enables the model to start relaxation at different noise levels, depending on the configuration of the unstable structure.
>
> Additionally, our analysis show that adaptive scheduling based on the diffusion time prediction is beneficial as it allows the model to flexibly adjust the process (i.e. jump back and forth in time) which leads to significantly improved performance of the two flexible models (MoreRed-JT and MoreRed-AS) over the model where only the starting point is predicted (MoreRed-ITP). We added a figure to show some example trajectories of predicted time steps over relaxation that visualize this flexibility (see Fig. 2b and Section 4.1). This is also where the applicability of the time step prediction to other diffusion-based frameworks comes from: We show that having a flexible reverse diffusion process might not only be beneficial when starting from non-Gaussian input such as unstable molecules but also for data generation from scratch, e.g. when sampling molecules or images from Gaussian noise (see Tables 1 and 2).
>
> Therefore, the time step predictor is both an integral part of MoreRed and a promising contribution to other fields. We argue that our approach is an innovative method for relaxation of molecules. In our evaluation on the QM7-X dataset, we demonstrate  its usefulness in a thorough study where we show that it is a viable alternative to ML force field models that does not require the expensive training data with unstable structures with force labels, but still learns to map onto the data manifold of stable molecules.

---

> > ### Author Response · Authors · 2023-11-22
> > **Answer 2/2 to Reviewer 2WpS**
> >
> > > **Questions:**
> > >
> > > The paper uses PaiNN as the backbone and compares it with force field (FF) methods also based on PaiNN. I would like to see the authors try using other different backbones to evaluate the robustness of their proposed method concerning the choice of backbone.
> >
> > Thank you for bringing this up and we agree that it is a valuable experiment to strengthen our work. Therefore, we conducted a new set of experiments using the SO3net architecture as implemented in SchNetPack  [1] - a spherical harmonics based neural network in the spirit of Tensor Field Networks [2] and NequIP [3] - as a backbone representation in our framework. Our findings are summarized in Appendix A.4.4 and Fig. A7. Overall, we can confirm that our approach works similarly well using this different equivariant neural network backbone and still outperforms the ML force field regarding the structure accuracy. The general performance for all models, including the force field, is slightly worse than with PaiNN and there is a slight difference between the performance of MoreRed-AS and MoreRed-JT. We believe that this comes from the fact that, due to the time limitation, hyperparameter tuning was not feasible and we had to use half the number of parameters employed in PaiNN to speed-up the training and relaxation experiments. Despite this constraint, we maintained fairness by using identical model hyperparameters for both MoreRed and the force field. Moreover, we would like to mention that these constraints were due to the long training of the force field model, which uses 100 times more data than MoreRed and needs more than 7 days to converge when using large models with optimal parameters. In contrast, MoreRed requires solely stable and unlabeled structures for training and converges in less than two days. This contrast reinforces the efficiency of our approach.
> >
> > We would like to thank you again for your valuable time. If you feel that your questions and concerns have been addressed appropriately, we would be glad about an increased rating.
> >
> > **Citations:**
> >
> > [1] Schütt, Kristof T., et al. "SchNetPack 2.0: A neural network toolbox for atomistic machine learning." The Journal of Chemical Physics 158.14 (2023).
> >
> > [2] Thomas, Nathaniel, et al. "Tensor field networks: Rotation-and translation-equivariant neural networks for 3d point clouds." arXiv preprint arXiv:1802.08219 (2018).
> >
> > [3] Batzner, Simon, et al. "E (3)-equivariant graph neural networks for data-efficient and accurate interatomic potentials." Nature communications 13.1 (2022): 2453.

---

> > > ### Comment · Reviewer_2WpS · 2023-11-23
> > > **Thanks the authors' reply**
> > >
> > > Thank you for your response, authors.
> > >
> > > Regarding the supplementary experiments to further validate the generality of using alternative backbones, I appreciate your effort in conducting these experiments within the limited time. However, I would have liked to see more diverse backbone choices. Since today is the last day of discussion, I understand that it might not be feasible to include additional experiments. For now, I consider this issue to be resolved.
> > >
> > > However, I still have some concerns about the contribution of "Diffusion Time Step Prediction" that the authors proposed. In the paper, the authors mentioned the following viewpoint:
> > > > To perform reverse diffusion starting from unstable molecules at arbitrary noise levels (i.e. not sampled from the noise distribution $p_T\left(\mathbf{x}\_T\right)$ ), it is necessary to set the initial diffusion time step appropriately. More specifically, we need to know from which diffused distribution $p_t$, the input molecule $\tilde{\mathbf{x}}$ was drawn. It is necessary to estimate an appropriate diffusion time step $\hat{t}$ such that $p_{\hat{t}}(\tilde{\mathbf{x}})$ is sufficiently high. Therefore, we propose a time step predictor in the next subsection.
> > >
> > > I acknowledge this viewpoint, as it is true that the input molecule structures provided to the model may not necessarily be sampled from $p_{T}$. However, for image generation, this issue does not exist since we sample noise from $p_{T}$ and progressively denoise it. Therefore, I find the motivation for its use in image generation less convincing.
> > >
> > > In summary, I appreciate the authors' efforts and responses (although it is a bit late, and there is limited time left for discussion). Despite some lingering questions, I am willing to revise my score. I will temporarily increase my score to 6.

---

> > > > ### Author Response · Authors · 2023-11-23
> > > >
> > > > > Regarding the supplementary experiments to further validate the generality of using alternative backbones, I appreciate your effort in conducting these experiments within the limited time. However, I would have liked to see more diverse backbone choices. Since today is the last day of discussion, I understand that it might not be feasible to include additional experiments. For now, I consider this issue to be resolved.
> > > >
> > > > Thank you for your swift response, and we are glad that this addressed concern was resolved. While we would have preferred to conduct additional experiments with diverse backbones, the integration of models from other packages into our code, along with the necessary validation, training, and experimentation, exceeds the time constraints of the rebuttal period. However, we commit to exploring this aspect in our future work.
> > > >
> > > > > However, I still have some concerns about the contribution of "Diffusion Time Step Prediction" that the authors proposed. In the paper, the authors mentioned the following viewpoint:
> > > > >
> > > > > > To perform reverse diffusion starting from unstable molecules at arbitrary noise levels (i.e. not sampled from the noise distribution $p_Tx_T$ ), it is necessary to set the initial diffusion time step appropriately. More specifically, we need to know from which diffused distribution $p_t$, the input molecule $\tilde{\mathbf{x}}$ was drawn. It is necessary to estimate an appropriate diffusion time step $\hat{t}$ such that $p_{\hat{t}}(\tilde{\mathbf{x}})$ is sufficiently high. Therefore, we propose a time step predictor in the next subsection.
> > > > >
> > > > > I acknowledge this viewpoint, as it is true that the input molecule structures provided to the model may not necessarily be sampled from $p_{T}$. However, for image generation, this issue does not exist since we sample noise from $p_{T}$ and progressively denoise it. Therefore, I find the motivation for its use in image generation less convincing.
> > > >
> > > > We acknowledge your concern regarding the motivation for utilizing time step prediction for its use in image generation. To tackle your concerns, we want to give two arguments.
> > > >
> > > > First, we find that also in images it may potentially be very helpful to sample from other distributions than $p_{T}$. This could for example be the case in image restoration, where tasks like super-resolution, deblurring, inpainting, and colorization are performed on input images [4]. The restoration process uses a very similar concept as the task of optimizing molecules with diffusion models. We can view the initial images before restoration as data points with an arbitrary level of noise and want to optimize them in such a way that the closest point on the data manifold (similar to the stable structure) is obtained. In order to estimate the distance to the data manifold it is necessary to find a good starting point $\hat{t}$ in the diffusion process. Due to the limited time of the rebuttal, we will not be able to perform any experiments on this, but certainly we will address this task in future work.
> > > >
> > > > Furthermore, even in ordinary image generation, where we sample noise from $p_{T}$, errors can occur in the reverse diffusion process. By adapting $\hat{t}$ according to the current state of the sample (as predicted by the time step predictor), a flexible reverse diffusion scheme emerges, which can correct for such errors. In the paper we write:
> > > > >  We observe that MoreRed, i.e. adaptive scheduling, performs better than DDPM, i.e. fixed scheduling, in all criteria. [...] This confirms our hypothesis that the adaptive reverse diffusion procedure based on the time step prediction is beneficial for unconditional sampling from Gaussian noise. MoreRed can adapt the time step, as can be seen in some exemplary sampling trajectories in Figure A8, to correct for errors. We further discuss these findings in Appendix A.4.5.
> > > >
> > > > Besides, we would like to refer to the ‘general answer 2 to all reviewers’ in the top of the rebuttal, where we further discuss why the time step prediction and the idea of using an adaptive schedule improves the unconditional data generation, including images.
> > > >
> > > > > In summary, I appreciate the authors' efforts and responses (although it is a bit late, and there is limited time left for discussion). Despite some lingering questions, I am willing to revise my score. I will temporarily increase my score to 6.
> > > >
> > > > We apologize for the delayed response. Preparing the extensive additional experiments and results, coupled with the paper revision and the detailed answers for all reviewers' concerns and questions required a lot of time and effort. We appreciate the score increase and your understanding.
> > > >
> > > > **Citations**:
> > > >
> > > > [4] Kawar, Bahjat, et al. "Denoising diffusion restoration models." Advances in Neural Information Processing Systems 35 (2022): 23593-23606.

---

> > > > > ### Comment · Reviewer_2WpS · 2023-11-23
> > > > > **Thanks for your further clarification**
> > > > >
> > > > > Thank you for the further explanation, authors.
> > > > >
> > > > > Your response has effectively addressed my concerns, and I would be willing to raise my score to 8 if there is experimental support. However, time constraints are evidently not allowing for that. Additionally, I look forward to seeing whether your 'Diffusion Time Step Prediction' can be extended to discrete diffusion.
> > > > >
> > > > > Thank you again for your response, and I look forward to an improved version of your paper in the future!

---

### Official Review · Reviewer_jJY4 · 2023-10-27

**Soundness:** 2 fair
**Presentation:** 3 good
**Contribution:** 1 poor
**Rating:** 5
**Confidence:** 4

**Summary:**

The authors propose a diffusion model to find stable molecular structures. When using it, instead of starting with a normal distribution, it becomes a reverse diffusion process from any unstable structure to a stable structure. Therefore, the critical technical contribution is to use a network that determines the corresponding diffusion time step for the input unstable structure.
In the experiments on QM7-X, the proposed method outperforms L-BFGS using a force field in thems of relaxed structure reproduction.
In addition, they show that diffusion time step prediction is empirically effective for molecule and image generation.

**Strengths:**

* The proposed method can learn only from stable structures, so the required training data is much smaller than ML force fields. I think this advantage is significant when the training data is collected by a more accurate but heavy QM method, such as CCSD(T).
* The diffusion time step prediction is a novel trick for reverse diffusion from non-gaussian input.
* The readers can see the feasibility of the diffusion time step prediction via the experimental results in Figure 2 (Right).

**Weaknesses:**

* The application of the proposed method needs to be described so that the impact of this method is not clear.   I would like to see how the proposed molecule relaxation can be used in chemistry or biology.
* The proposed method is 10 times slower than the structure optimization using a force field model.
* It needs to be explained why the RMSD ratio, the RMSD after relaxation divided by the RMSD of the unstable initial structure, is used for the comparison. I am worried that the proposed method can be only accurate for initial structures near their relaxed structures.
* No explanation why the diffusion time step prediction is also helpful for unconditional generation from normal distribution prior.

**Questions:**

* What is the application of molecule relaxation without energy/force prediction? I guess you have some assumptions about applying the proposed method, but it is not described in the main text.
* How long does it take to find the relaxed structure from an initial structure? Is 0.05s * 1000 (MoreRed-AS) vs 0.03s * 118 (FF model) correct?
* Could you provide RMSD instead of RMSD ratio for Figure 3a?
* Is it possible to output the energy or force of stable structures? It extends the applications of this method, such as crystal structure predictions. If the energy prediction head is added to the proposed method, Matbench Discovery[1] can be a good platform for evaluating the proposed method.

[1] https://matbench-discovery.materialsproject.org/ , https://arxiv.org/abs/2308.14920

**Details Of Ethics Concerns:**

No concern.

---

> ### Author Response · Authors · 2023-11-22
> **Answer 1/3 to Reviewer jJY4**
>
> Dear reviewer jJY4,
>
> thank you for committing your valuable time to provide us with helpful, constructive feedback. In the following, we have addressed your concerns in a point-by-point answer.
>
> > **Weaknesses:**
> >
> > - The application of the proposed method needs to be described so that the impact of this method is not clear. I would like to see how the proposed molecule relaxation can be used in chemistry or biology.
>
> Thank you for pointing this out. We updated the Introduction to reflect the possible impact of our framework more clearly. Our main target application is to find stable molecule structures by relaxing unstable structures proposed by cheap methods, e.g., with empirical MMFF, and our MoreRed can complement the well-established force field (FF) models in the domain where unstable structures with force labels for training FF models are not available or hard to obtain, e.g. when using very accurate but costly QM methods or when working with large structures. This could, for example, have a huge impact on crystal structure search, where large databases like MaterialsProject [2] exist, but only provide stable structures with labels. In such cases, all FF approaches cannot be used due to the lack of unstable structures. In our studies, we thoroughly evaluate our method on the QM7-X dataset as it allows us to train both MoreRed and an ML force field for a proper comparison. Our main contribution is that we propose a novel approach to perform structure optimization based on learning a pseudo potential energy surface using solely stable structures. While achieving promising results in systems in the scale of QM7-X opens a new promising path, extending diffusion models to handle periodic boundary conditions or very large molecules still is an open challenge in itself, and thus remains a task for future work [3].
>
> > - The proposed method is 10 times slower than the structure optimization using a force field model.
>
> This is partially true. We reported an average inference time per step for a batch of 128 structures of 0.03 s for the force field and 0.05 s for the MoreRed variants and a median (not mean because the distribution of the number of steps is highly skewed) number of relaxation steps until convergence of 118 for the FF model, 53, 219, and 1000 for MoreRed-ITP, MoreRed-JT, and MoreRed-AS, respectively. Hence, it is true that using the best performing variant MoreRed-AS takes longer in the median. However, the simplest variant that only predicts the starting $t$ (MoreRed-ITP) requires less steps than the ML force field in median while still outperforming it with regard to the RMSD ratio (see Figure 3a), and the MoreRed-JT variant, which outperforms the force field model and requires only two times more steps in median. On top of that, we find that all MoreRed variants can easily be parallelized during sampling and run the whole denoising on the GPU, while parallelization of the force field with the L-BFGS optimizer can be non-trivial. The parallelized optimization of 20k structures in our experiments took about 3 hours for MoreRed-AS on NVIDIA A100 while the sequential optimization with the force field model took about 49 hours utilizing the same hardware requirements and using L-BFGS from the ASE library, according to the log files. Because of the added overhead of loading from and to the GPU and converting to and from ASE classes after each relaxation iteration using L-BFGS in ASE, the running time increases further. We will give a more detailed theoretical calculation on the computation time for structure optimization in the questions section below as asked.

---

> > ### Author Response · Authors · 2023-11-22
> > **Answer 2/3 to Reviewer jJY4**
> >
> > > - It needs to be explained why the RMSD ratio, the RMSD after relaxation divided by the RMSD of the unstable initial structure, is used for the comparison. I am worried that the proposed method can be only accurate for initial structures near their relaxed structures.
> >
> > We agree that this might be confusing. We decided to use the RMSD ratio as it is a concise way to express whether the relaxation brought the structure of interest closer to its stable state or not (cases with ratio > 1 have not succeeded in getting closer to the stable state and we refer to these as outliers or failing cases). Besides, when reporting only the RMSD, it could be hard to interpret because it is not a normalized measure and it can happen that one model shows less RMSD level but fails in more cases as one can see in the higher outlier percentage for the force field model than MoreRed variants. However, in order to make our results more transparent and convincing, we added several plots in Fig. A6 which shows the RMSD error (not the ratio) and updated the manuscript to discuss these results in Appendix A.4.3. It can be seen that MoreRed performs well on both structures initially close to or far away from their relaxed structure. Only the simplest variant, which predicts only the starting time step and applies reverse diffusion with fixed scheduling (MoreRed-ITP), shows impaired performance for unstable QM7-X structures with comparatively larger RMSD. This can be explained by the restricted flexibility of this variant. The adaptive scheduling of the other two variants allows them to dynamically steer the relaxation and push the sample closer to the target distribution by moving back and forth in time to account for its errors (see newly added Fig. 2b with trajectories of predicted time steps). Hence, these two variants are clearly more successful in relaxing the test structures with large RMSD. This reinforces the significance of the time step prediction for our approach and we have updated Section 4.1 to better reflect this.
> >
> > > - No explanation why the diffusion time step prediction is also helpful for unconditional generation from normal distribution prior.
> >
> > Thank you for bringing this up. As mentioned in our previous point, the MoreRed variants with adaptive scheduling achieve the best results. This comes from their flexibility in running longer/shorter relaxation with adapted time steps. They can account for errors resulting from the stochasticity of the noise prediction by going back in time or skip to lower time steps if a relaxation step brings them close to the data manifold (as can be seen in the newly added Fig. 2b and Fig. A8 left with trajectories of predicted time steps). We also added a more detailed general response 2 above for all reviewers. Besides, we have added an explanation to Section 4.3 in the paper and in details in the Appendix A.4.5 about  why the time step predictor is also promising for unconditional data generation.
> >
> > > **Questions:**
> > >
> > > - What is the application of molecule relaxation without energy/force prediction? I guess you have some assumptions about applying the proposed method, but it is not described in the main text.
> >
> > As mentioned in our first point above, our target application is to find stable molecule structures by relaxing unstable structures proposed by cheap methods. Users of course want to know the physical property of the obtained stable structures.  To this end, one can for example use QM methods applied only to the stable molecules, which is much cheaper than those applied throughout the whole relaxation process.  Alternatively, one can train an additional output head or separate model to predict physical properties, which can  also be  trained only on stable structures.

---

> > > ### Author Response · Authors · 2023-11-22
> > > **Answer 3/3 to Reviewer jJY4**
> > >
> > > > - How long does it take to find the relaxed structure from an initial structure? Is 0.05s * 1000 (MoreRed-AS) vs 0.03s * 118 (FF model) correct?
> > >
> > > Computing the relaxation times is not as straightforward as suggested. As mentioned above, the median number of steps until convergence strongly depends on the MoreRed variant, ranging from 53 steps for the fastest MoreRed-ITP until 1000 steps for the most accurate MoreRed-AS. Please note that all variants were able to outperform the force field model on the RMSD ratio, which means that one can use MoreRed-JT for example, which takes only twice more steps in median than the force field. Furthermore we have to distinguish between sequential and parallelized optimization of structures:
> > >
> > > **Sequential Optimization:**
> > > Comparing MoreRed variants to the force field model, we observed an average inference time for a single structure, not a batch, of 0.03s for MoreRed and 0.02s for the force field model. To compute the mean inference time per structure optimization performed sequentially, we need to use not the median but the mean. The mean number of optimization steps until convergence was measured as 64 steps for MoreRed-ITP, 489 steps for MoreRed-JT,  992 steps for MoreRed-AS, and 122 steps for the force field model.  Therefore, the average total inference time per structure optimization are 0.03s * 64 = 1.92s for MoreRed-ITP, 14.67s for MoreRed-JT, 29.76s for MoreRed-As and 2.44s for the force field.
> > >
> > > **Parallelized Optimization:**
> > > For efficient optimization of a large number of structures, as it is usually the case in many applications, model inference is preferably done in batches. While this can be done straight forward for all variants of MoreRed, parallelizing structure optimization with force fields is not a trivial task. This is mainly related to the need of the L-BFGS optimizer, which does not exist in a parallelizable implementation for our case of use and for our knowledge.   Nevertheless, assuming batchwise relaxation with the force field is possible, we observed an inference time for evaluating a batch of 128 molecules of 0.03s and 0.05s for the force field and MoreRed variants, respectively. Since the batchwise relaxation is done until all the structures in the batch have converged, in a worst case scenario, which is more likely to happen the larger the batch is, both methods need the maximum number of allowed steps, 1000. This would result in a total inference time per structure of (0.03s * 1000)/128 = 0.23s for the force field and (0.05s * 1000)/128 = 0.39s for MoreRed variants. Making MoreRed only 1.7 times slower.
> > >
> > > Since we did not clarify this in the main text, we added an additional section on the computation time in Appendix A.4.6.
> > >
> > > > - Can you provide RMSD instead of RMSD ratio for Figure 3a?
> > >
> > > We have added Fig. A6 where we report the RMSD and additional results to further examine the performance and behavior of our three MoreRed variants. The results are similar to the RMSD ratio in that all variants perform similar to or better than the ML force field on QM7-X. However, we see that the simplest variant that only predicts the starting time step for relaxation (MoreRed-ITP) shows impaired performance for structures that have a large RMSD (see Fig. A6c). Our analysis suggests that this might be caused by the fixed time schedule.MoreRed-AS/-JT equipped with adaptive scheduling are more successful in relaxing these structures at the expense of more diffusion iterations (see Fig. 2b).
> > >
> > > > - Is it possible to output the energy or force of stable structures? It extends the applications of this method, such as crystal structure predictions. If the energy prediction head is added to the proposed method, Matbench Discovery[1] can be a good platform for evaluating the proposed method.
> > >
> > > Yes it is possible to add an energy prediction head to the backbone representation, similar to the prediction of the time step, which are both invariant quantities. Note that we are using the same backbone used to train the force field model, which predicts the energy as well. However the Matbench Discovery dataset can currently not be used for evaluation, because our method focuses on the optimization of molecules, not materials and on the accuracy of the optimized structure. Extending diffusion models to handle periodic boundary conditions still is an open challenge in itself [3] and remains as a task for future work that can open potential paths.
> > >
> > > We would like to thank you again for your valuable time. If you feel that your questions and concerns have been addressed appropriately, we would be glad about an increased rating.
> > >
> > > **Citations:**
> > >
> > > [2] Jain, Anubhav, et al. "Commentary: The Materials Project: A materials genome approach to accelerating materials innovation." APL materials 1.1 (2013).
> > >
> > > [3] Yang, Mengjiao, et al. "Scalable Diffusion for Materials Generation." arXiv preprint arXiv:2311.09235 (2023).

---

> > > > ### Comment · Reviewer_jJY4 · 2023-11-23
> > > >
> > > > Thank you for the very detailed response and additional experimental results.
> > > >
> > > > > To this end, one can for example use QM methods applied only to the stable molecules, which is much cheaper than those applied throughout the whole relaxation process.
> > > >
> > > > I agree that the proxy of QM methods in structure optimization can be an application of the proposed method.
> > > > I recommend citing references in which a structure optimization using the MLFF is employed before QM methods.
> > > >
> > > > Now, I would like to know how the reported RMSDs affect the subsequent analysis using QM methods.  For a simple example, I would like to know the DFT energy reproducibility of relaxed structures obtained by the proposed method and the MLFF.

---

> > > > > ### Author Response · Authors · 2023-11-23
> > > > >
> > > > > Dear reviewer jJY4,
> > > > >
> > > > > thank you for your timely response. We are glad that you appreciate our additional experiments and answers.
> > > > >
> > > > >
> > > > > > I agree that the proxy of QM methods in structure optimization can be an application of the proposed method.
> > > > > > I recommend citing references in which a structure optimization using the MLFF is employed before QM methods.
> > > > >
> > > > > Thank you, we will update our related work with corresponding citations for the final version.
> > > > > > Now, I would like to know how the reported RMSDs affect the subsequent analysis using QM methods. For a simple example, I would like to know the DFT energy reproducibility of relaxed structures obtained by the proposed method and the MLFF.
> > > > >
> > > > > We strongly agree that these are very interesting research questions. We plan to conduct additional experiments in the future, where we apply MoreRed to other data and utilize DFT to analyze and verify the obtained structures, e.g. with respect to the energy, and we are thankful for your thoughts on this.
> > > > > For our QM7-X experiments, an evaluation with DFT is difficult. The dataset was assembled with a method from FHI-aims, which is proprietary and not accessible for us. Using a different QM method will always lead to slightly changed energy surfaces and therefore introduce additional errors. Accordingly, it is not clear how sensible and fair an analysis with a different QM implementation would be. Therefore, we decided to assess the performance of our method based on the RMSD to the structures reported in QM7-X, which allows a thorough comparison between the MLFF and our method.
> > > > > We have evaluated the energy of molecules relaxed with MoreRed using the MLFF before. There we saw that the predicted energy was only marginally higher than for the structures optimized with the FF. Of course, we are fully aware that this is, first, a shallow comparison in favor of the MLFF and, second, no replacement for analysis with DFT.

---

### Official Review · Reviewer_vPT3 · 2023-10-30

**Soundness:** 2 fair
**Presentation:** 3 good
**Contribution:** 2 fair
**Rating:** 3
**Confidence:** 3

**Summary:**

This paper proposes to formulate molecular relaxation as a statistical learning task, defining a diffusion process that goes from unstable molecules to stable ones. This is in contrast to most state-of-the-art techniques that try to imitate the physical forces that drive this process in nature.

**Strengths:**

- Using a diffusion process for molecular relaxation is novel and creative.
- The problem is described in a very easy-to-understand and intuitive manner and the connection to diffusion modeling is well made.
- The authors have written a great background on diffusion modeling.

**Weaknesses:**

- The parameterization of the noise process is simple Gaussian blurring. It is not clear if this is a realistic assumption. It might be the case that this method wrongly describes molecules as stable when they are unstable, just because it is unable to identify the presence of Gaussian blurring.
- In the second paragraph of page 8, the authors mention that adding their synthetic noise on stable molecules causes force field methods to not be able to optimize the molecules to their stable starting point, whereas their diffusion model is able to perform this. However, this is not a valid benchmark, since the force field methods are supposed to predict forces on physically plausible molecules whereas the diffusion process that the authors have designed makes no guarantees about the plausibility of the molecules. This experiment can be reframed to show a weakness of the proposed method, in which unphysical starting points still become "stable", which should not be the case.
- Similar to the criticism above, I don't believe that the time step prediction performance as reported in section 4.1 is relevant as independent Gaussian noise on the atoms is somewhat easier to predict than stable vs unstable molecules. Since QM7-X contains unstable structures as well, a better approach would be to see the correlation between the time step predictions of the network versus the ground truth RMSDs.
- The test runs should be done with a few different random seeds so that we see how much of a discrepancy exists between runs for this method, as it is statistical in nature.

**Questions:**

- How does the time step prediction compare to non-fixed variance approaches such as [1] and [2]? It seems very similar to the approach taken there, why not mention how your work is different?
- Dataset scaling is frequently mentioned as a benefit to this method, although it seems that with larger datasets, the probability that some of the modes of the mixture of Gaussians you use as your prior distribution would get mixed. That is, once you add some noise to some training structure $y_i = x_i + \epsilon$, then you might have the issue that for some $j \neq i$, $||y_i - x_i|| > ||y_i - x_j||$. Is this a problem?
- For table 1, there are many methods that generate molecules on QM9, why not report some of those results as well?

Citations

[1] Alex Nichol and Prafulla Dhariwal. Improved denoising diffusion probabilistic models. arXiv:2102.09672, 2021.

[2] Prafulla Dhariwal and Alex Nichol. Diffusion Models Beat GANs on Image Synthesis. arXiv:2105.05233, 2021.

---

> ### Author Response · Authors · 2023-11-22
> **Answer 1/3 to Reviewer vPT3**
>
> Dear reviewer vPT3,
>
> thank you for taking your valuable time and providing us with valuable feedback. In the following, we have addressed your concerns and questions point-by-point. If you feel that we have addressed your concerns appropriately, we would be happy about an increased rating.
>
> > **Weaknesses:**
> >
> > - The parameterization of the noise process is simple Gaussian blurring. It is not clear if this is a realistic assumption. It might be the case that this method wrongly describes molecules as stable when they are unstable, just because it is unable to identify the presence of Gaussian blurring.
>
> Although the diffusion model  is trained with Gaussian blurred samples, it does not learn the Gaussian noise, but the derivative of the log probability or score function  (see Eq.(1)).   Therefore, it does not converge to an unstable point where no training points exist in the neighborhood, even if it does not contain Gaussian noise. However, it is indeed sensible to worry that our diffusion time predictor could learn to detect Gaussian noise (e.g., independent randomness or Gaussianity), instead of detecting the distance to the data manifold.  Following your suggestion in another point, we added the new Fig. 2a in the revision, which denies this worry.  Namely, the predicted diffusion time of the unstable molecules in QM7-X, which is NOT Gaussian blurred, has high correlation to the distance (RMSD) to their stable molecules, and the network never predicts time step 0 for any unstable molecules in QM7-X.  This implies that the diffusion time predictor does not rely on measuring Gaussian noise but measuring the distance to the data manifold by identifying specific correlations that occur in the stable structures in the training data. Our experiments in Fig. 3a also supports this, i.e. when we relax the unstable QM7-X structures, which were not Gaussian blurred,  MoreRed relaxes them as successfully as the ML force fields do. Furthermore, 94.5% of the molecules obtained in our unconditional data generation experiments with the time step prediction are physically valid, i.e. not just any structures without Gaussian noise (see Table 1). We appreciate the reviewer’s insightful suggestion, which cleared this suspicion that general readers would have. Although training models with the force labels is a more physically grounded approach than using Gaussian blur, obtaining a sufficient amount of training force labels is computationally expensive.  Our main finding in this paper is that training a diffusion model, along with the diffusion time predictor, on Gaussian blurred, unlabeled stable structures can be a much cheaper alternative, requiring 100 times less data, but still capturing a physically meaningful data manifold of stable structures.

---

> > ### Author Response · Authors · 2023-11-22
> > **Answer 2/3 to Reviewer vPT3**
> >
> > > - In the second paragraph of page 8, the authors mention that adding their synthetic noise on stable molecules causes force field methods to not be able to optimize the molecules to their stable starting point, whereas their diffusion model is able to perform this. However, this is not a valid benchmark, since the force field methods are supposed to predict forces on physically plausible molecules whereas the diffusion process that the authors have designed makes no guarantees about the plausibility of the molecules. This experiment can be reframed to show a weakness of the proposed method, in which unphysical starting points still become "stable", which should not be the case.
> >
> > It is correct that our method exhibits a strong robustness with respect to the starting point of the relaxation. It is trained to converge to physically plausible molecules regardless of the physicality of the input. We believe that this robustness is a significant advantage, not a weakness, since it expands the space of the input where the method can be applied. In our assumed applications, the goal is to relax structures that are proposed by some cheap empirical force field. A robust model would allow one to freely choose the cheap method, e.g. using different implementations from different chemoinformatics packages. However, classical ML force fields tend to fail as soon as the starting point stems from a different method than used when assembling the training data set. This is one of their main weaknesses and it occurs because the input is out of distribution for the ML model (regardless of the physicality of the input). Therefore, we believe that our proposal of a more robust approach is a valuable contribution.
> >
> > For our application of relaxing structures obtained from cheap methods, we do not see any problem if MoreRed relaxes unphysical starting points to stable states. It would be helpful if the reviewer would further explain the situations in which this behavior is undesirable, so that we can discuss this point in the paper. If desired, one might add sanity checks to identify relaxations of unphysical starting points. For example, MoreRed is trained to map unphysical systems to physically plausible, stable systems. Accordingly, the RMSD of atom positions before and after relaxation will be comparatively large if a relaxation starts from an unphysical system, since it will involve a lot of movement of the atoms. Such sanity checks are cheap to compute and also often employed when running relaxations with DFT or with ML force fields (note that these methods also have no guarantee not to converge when the starting point is  unphysical).
> >
> > > - Similar to the criticism above, I don't believe that the time step prediction performance as reported in section 4.1 is relevant as independent Gaussian noise on the atoms is somewhat easier to predict than stable vs unstable molecules. Since QM7-X contains unstable structures as well, a better approach would be to see the correlation between the time step predictions of the network versus the ground truth RMSDs.
> >
> > Thank you for this idea. We added Fig. 2a where we examine the correlation between the RMSD of unstable structures in QM7-X and the predicted initial time step by our model.  It mainly shows two insights: There is a clear correlation between the RMSD and the predicted time step and the model always predicts time steps greater than zero for these unstable structures. There are few outliers in MoreRed-AS/-ITP where the model predicts larger time steps for structures with intermediate RMSD. However, we examine the performance of all three MoreRed variants with respect to the RMSD before and after relaxation in the newly added Fig. A6c and their performance is not impaired for structures with intermediate RMSD.
> >
> > > - The test runs should be done with a few different random seeds so that we see how much of a discrepancy exists between runs for this method, as it is statistical in nature.
> >
> > We agree, and have added Figure A6b to our paper to report the standard deviations across 5 test runs with different random seeds. Overall, the observed deviations are low, and do not affect our conclusions.

---

> > > ### Author Response · Authors · 2023-11-22
> > > **Answer 3/3 to Reviewer vPT3**
> > >
> > > > **Questions:**
> > > >
> > > > - How does the time step prediction compare to non-fixed variance approaches such as [1] and [2]? It seems very similar to the approach taken there, why not mention how your work is different?
> > >
> > > In [1,2], the conditional variance of $x_{t-1}$ given $x_t$ is learned for improving the test likelihood and data generation quality when the reverse diffusion steps are reduced.  Although we also use an additional model to predict the diffusion time step, which correlates to the noise component of $x_t$, our approach is fundamentally different from [1,2].  First, the conditional variance corresponds to the noise amplitude that a single reverse step should ADD after the noise removal, which corresponds to the last term in Eq.(3) in our paper, while our diffusion time predictor implicitly measures how much noise the whole remaining reverse steps should REMOVE from the current sample, i.e., $\|x_t - x_0\|^2$.  Apparently, the conditional variance prediction in [1,2] cannot be used to predict the diffusion time for setting the initial diffusion time for molecule relaxation nor for adaptive scheduling.  Second, [1,2] always use fixed scheduling, even if it is different from the training scheduling, while we use adaptive scheduling by using the diffusion time predictor.
> > >
> > > > - Dataset scaling is frequently mentioned as a benefit to this method, although it seems that with larger datasets, the probability that some of the modes of the mixture of Gaussians you use as your prior distribution would get mixed. That is, once you add some noise to some training structure y_i = x_i + epsilon , then you might have the issue that for some i != j, ||y_i - x_i|| > ||y_i - x_j|| . Is this a problem?
> > >
> > > The mixture components get overlapped when the training samples are dense or the noise level is large, and this degrades the diffusion time prediction performance for samples with large noise, as seen in Fig.A1 right (which was Fig.2 in the original submission).  However, the prediction for low noise samples is barely affected,  unless the dimension $D$ of the data space is very small.  Assume that there are two training molecules $x_a, x_b$ with the distance $r = \|x_a-x_b\|$, and consider Gaussian blurred samples of $x_a$ with the standard deviation equal to $\sigma = r$.  Although, in this situaion, $x_b$ lies in the high density shell of the Gaussian blurry of $x_a$ (the bump in the left plot in Fig A.1 left in the revision), the Gaussian blurred samples of $x_a$ are uniformly distributed in the still high $(D-1)$-dimensional shell, and therefore the probability that the Guassian blurry produces a sample close to $x_b$ is extremely low.  On the other hand, the diffusion time predictor is trained with a lot of samples from the neighborhood of $x_b$ as low noise samples from $x_b$ because the density is high.  Accordingly, the models are trained so that they recognize the molecules close to $x_b$ as low noise samples.  This intuition can be mathematically confirmed by computing the density ratio between the two mixture components around $x_b$, i.e., $Gauss_D(x_b + \epsilon; x_b, \delta^2) / Gauss_D(x_b + \epsilon; x_a, r^2)$ for $\|\epsilon\| ~ \delta << r$, which is extremely high when $D$ is not very small. We added this explanation in the discussion in Appendix A.2.1.
> > >
> > > > - For table 1, there are many methods that generate molecules on QM9, why not report some of those results as well?
> > >
> > > We added comparisons with E-NF [1], G-SchNet [2], EDM [3] and GeoLDM [4]  in Appendix A.4.5 and Table A1.  However, please note that the baselines and our MoreRed solve different problems. The baselines, which learn $p(X,Z)$ from the QM9 dataset, generate both atom types $Z$ and positions $X$, while our MoreRed, which learns $p(X|Z)$, generate atom positions given the atom types. We restricted our scope to the atom position generation because our main goal is to perform molecule relaxation.
> > >
> > > **Citations:**
> > >
> > > [1] Garcia Satorras, Victor, et al. "E (n) equivariant normalizing flows." Advances in Neural Information Processing Systems 34 (2021): 4181-4192.
> > >
> > > [2] Gebauer, Niklas, Michael Gastegger, and Kristof Schütt. "Symmetry-adapted generation of 3d point sets for the targeted discovery of molecules." Advances in neural information processing systems 32 (2019).
> > >
> > > [3] Hoogeboom, Emiel, et al. "Equivariant diffusion for molecule generation in 3d." International conference on machine learning. PMLR, 2022.
> > >
> > > [4] Xu, Minkai, et al. "Geometric latent diffusion models for 3d molecule generation." International Conference on Machine Learning. PMLR, 2023.

---

### Official Review · Reviewer_PyaS · 2023-10-31

**Soundness:** 2 fair
**Presentation:** 2 fair
**Contribution:** 2 fair
**Rating:** 5
**Confidence:** 3

**Summary:**

In this work, the authors studied a crucial problem in molecular modeling, molecule relaxation, via the methodology of generative modeling. Instead of learning a force field model to conduct simulation for molecule relaxation, the authors proposed MoreRed, which directly models the Boltzmann distribution of equilibrium molecular structures and also learns a diffusion step predictor for relaxation. Experiments are conducted to demonstrate the performance of MoreRed on molecular relaxation tasks.

**Strengths:**

1. The molecule relaxation task is of great interest to chemistry, biology and other scientific communities.
2. The proposed approach seems to be novel compared to previous approaches.

**Weaknesses:**

- **Regarding the evaluation settings**: As stated in Section 4.2, the authors use the QM7-X dataset to evaluate the molecule relaxation performance. The unstable molecules in this dataset were generated by sampling from the Boltzmann distribution of the stable molecules. The confidence of such data generation procedure is in doubt: (1) In real-world applications, given a molecule, we usually use either random conformation sampling or cheap conformation optimization (e.g., empirical MMFF) to obtain initial molecular structures. The authors should further clarify whether these generated unstable molecules in QM7-X match the real-world settings. (2) Beyond the small organic molecules, molecular relaxation is widely used to investigate the equilibrium state of molecular systems like the adsorbate-catalyst complex or protein-ligand complex. There exist large-scale benchmarks that are more related to real-world settings like Open Catalyst Project (IS2RE and IS2RS tasks). It would be more convincing to verify the proposed methods on these challenging tasks.
- **Regarding the generality of the proposed methods**: As a general framework, the proposed MoreRed can use different equivariant backbone models. Given the rich literature of equivariant networks, it would be more convincing to verify the generality of MoreRed with different architectures.
- **Regarding the compared baselines**: For molecule relaxation tasks, the authors only compare the proposed three variants of MoreRed with corresponding NN-based Force Field models on the RMSD metric. For the machine learning based molecule relaxation approaches, there indeed exists strong baselines like [1], and there also exists generative models capable of transforming one data distribution to another data distribution that also lie in the settings of the molecule relaxation tasks [2]. It would enhance the quality of this paper if the authors could provide further discussions and comparisons to these approaches.
- **Regarding the quality of unconditional generation**: I do not quite grasp how the diffusion time step predictor improve the unconditional generation quality. Moreover, the experiments also lack strong and advanced baselines [3,4,5].

[1] Lu, Shuqi, et al. "Highly Accurate Quantum Chemical Property Prediction with Uni-Mol+." arXiv preprint arXiv:2303.16982 (2023).

[2] Su, X., Song, J., Meng, C., & Ermon, S. (2022). Dual diffusion implicit bridges for image-to-image translation. arXiv preprint arXiv:2203.08382.

[3] Xu, M., Yu, L., Song, Y., Shi, C., Ermon, S., & Tang, J. (2022). Geodiff: A geometric diffusion model for molecular conformation generation. arXiv preprint arXiv:2203.02923.

[4] Jing, Bowen, et al. "Torsional diffusion for molecular conformer generation." Advances in Neural Information Processing Systems 35 (2022): 24240-24253.

[5] Xu, M., Powers, A. S., Dror, R. O., Ermon, S., & Leskovec, J. (2023, July). Geometric latent diffusion models for 3d molecule generation. In International Conference on Machine Learning (pp. 38592-38610). PMLR.

**Questions:**

Please see the comments in the Weaknesses section.

---

> ### Author Response · Authors · 2023-11-22
> **Answer 1/3 to Reviewer PyaS**
>
> Dear reviewer PyaS,
>
> thank you for taking your precious time and providing us with valuable feedback. In the following, we have addressed your concerns and questions point-by-point. If you feel that we have addressed your concerns appropriately, we would be happy about an increased rating.
>
> > **Weaknesses:**
> >
> > - **Regarding the evaluation settings:** As stated in Section 4.2, the authors use the QM7-X dataset to evaluate the molecule relaxation performance. The unstable molecules in this dataset were generated by sampling from the Boltzmann distribution of the stable molecules. The confidence of such data generation procedure is in doubt: (1) In real-world applications, given a molecule, we usually use either random conformation sampling or cheap conformation optimization (e.g., empirical MMFF) to obtain initial molecular structures. The authors should further clarify whether these generated unstable molecules in QM7-X match the real-world settings. (2) Beyond the small organic molecules, molecular relaxation is widely used to investigate the equilibrium state of molecular systems like the adsorbate-catalyst complex or protein-ligand complex. There exist large-scale benchmarks that are more related to real-world settings like Open Catalyst Project (IS2RE and IS2RS tasks). It would be more convincing to verify the proposed methods on these challenging tasks.
>
> We agree that the distribution of the initial unstable molecules in the real-world setting does not follow the Boltzmann distribution, and in this sense, QM7-X is maybe not the optimal data set for evaluation.  However, due to the high computational cost of generating datasets for chemical structures, only a limited number of these datasets is available to use. For a fair comparison of our method to machine learning force fields on the task of structure optimization, we require a dataset that provides a large number of stable/equilibrium structures with various compositions, as well as corresponding unstable/non-equilibrium variants of these with labels for energy and forces. Although the QM7-X dataset has some disadvantages, it is to the best of our knowledge the only molecule dataset fulfilling these requirements. For this reason it is often used to test the performance of machine learning force field models [6-8], which were proposed for the same real-world applications. Besides, it is common practice to utilize datasets in the scale of QM7-X like QM9, QM7 and MD17 to validate novel machine learning force fields [7-14].
>
> The stable structures of QM7-X are generated in exactly the same way you described. First,  a set of sufficiently different initial 3D structures were obtained using the MMFF94 force field for each SMILE string to cover many conformations. Then, all structures were subsequently optimized with accurate DFT simulations. The unstable structures were generated by displacing each molecular structure along a linear combination of normal mode coordinates computed at the DFTB3+MBD level within the harmonic approximation. Based on the equipartition theorem from classical statistical mechanics, the normal modes for the displacements were chosen in such a way that the energy difference between the unstable and stable structures follow a Boltzmann distribution. In their paper, the authors of the QM7-X dataset argue theoretically and provide analysis showing that the resulting structures are physically relevant  and cover the potential energy surface.
>
> The distribution of the unstable structures in QM7-X is indeed different from the real-world distribution, where the candidates are obtained with cheap optimization methods. However, both give physically feasible unstable structures that cover the potential energy surface with non-Gaussian noise, and therefore we expect that the results are still indicative for the realistic scenario. In the revision, we added a discussion on the choice and shortcomings of QM7-X (see Appendix A.4.1).
>
> While large scale benchmarks like the Open Catalyst Project would provide a challenging task to test structure optimization methods, it is a special use case of molecular relaxation, which requires the model to handle structures with periodic boundary conditions, and therefore is out of the scope of our contribution. Extending diffusion models and generative models for atomistic systems to handle periodic boundary conditions is an interesting open challenge in itself [17] and we leave it as future work.

---

> > ### Author Response · Authors · 2023-11-22
> > **Answer 2/3 to Reviewer PyaS**
> >
> > > - **Regarding the generality of the proposed methods:** As a general framework, the proposed MoreRed can use different equivariant backbone models. Given the rich literature of equivariant networks, it would be more convincing to verify the generality of MoreRed with different architectures.
> >
> > Thank you for bringing this up. We agree that it is a valuable experiment to strengthen our work. We conducted a new set of experiments using SO3net [16] as a backbone representation in our framework. This representation incorporates spherical harmonics to handle SO(3)-equivariance, distinguishing it from the PaiNN architecture.
> >
> > Our findings are summarized in Appendix A4.4 and Fig. A7. Overall, our MoreRed approach works similarly well using this different equivariant neural network backbone and still outperforms the ML force field in terms of  the structure accuracy. The general performance with SO3net is slightly worse than with PaiNN and there is a slight difference between the performance of MoreRed-AS and MoreRed-JT. This likely comes from the fact that, due to the time limitation by the rebuttal deadline, hyperparameter tuning was not feasible and we had to use half the number of parameters employed in PaiNN to speed-up the training and relaxation experiments. Despite this constraint, we maintained fairness by using identical model hyperparameters for both MoreRed and the force field. Moreover, we would like to mention that these constraints were due to the long training time of the force field model, which uses 100 times more data than MoreRed and needs more than 7 days to converge when using large models with optimal parameters. In contrast, MoreRed requires solely stable structures for training and converges in less than two days. This contrast reinforces the efficiency of our approach.
> >
> > > - **Regarding the compared baselines:** For molecule relaxation tasks, the authors only compare the proposed three variants of MoreRed with corresponding NN-based Force Field models on the RMSD metric. For the machine learning based molecule relaxation approaches, there indeed exists strong baselines like [1], and there also exists generative models capable of transforming one data distribution to another data distribution that also lie in the settings of the molecule relaxation tasks [2]. It would enhance the quality of this paper if the authors could provide further discussions and comparisons to these approaches.
> >
> > Thank you for pointing this out. We were not aware of Uni-Mol+ and we included it in the related work section. We can clearly see some similarities in the methodology, however, the task solved is different. Abstractly,  Uni-Mol+ as well as MoreRed are trained by denoising augmented trajectories of artificial data starting from stable structures. Besides the similar approach of augmenting training data by building trajectories between stable and unstable structures, we identify two main differences. First of all, Uni-Mol+ only adds minor perturbations for sampling the unstable structures, such that the structures remain close to their local minima, whereas MoreRed diffuses the stable structures to a distribution of Gaussian noise. More importantly, as a second key difference, Uni-Mol+ is designed for predicting properties computed for stable molecules using the slightly perturbed initial structures resulting from cheap MMFF methods as input. Accordingly, Uni-Mol+ is benchmarked on the property prediction tasks like initial structure to relaxation energy, where it achieves remarkable results. The publication reports, for instance, the MAE of the relaxation energy. However, it does not evaluate the quality of the predicted stable structures. The MAE might still be accurate even if the relaxed structure is slightly noisy. In our paper, we focus on the accuracy of the relaxed structure with respect to the true stable structure, which is why we focus on the RMSD measure as used in previous work [8, 14, 15]. We do not see Uni-Mol+ as a baseline in this task, since it was not designed for nor evaluated on it before. Nevertheless, employing our model for the property prediction task by adding an additional output head is an interesting avenue and we plan to explore this in the future. There, Uni-Mol+ will indeed be a strong baseline.
> >
> > Paper [2] targets the problem of translating from one distribution to another one by bridging between two different diffusion models.  We are not sure how this can be comparable to our method or how this would be used in the case of molecule relaxation, but we would be happy if you could elaborate more on this idea.
> >
> > Again, we are thankful for these interesting insights and we added them to our related work.

---

> > > ### Author Response · Authors · 2023-11-22
> > > **Answer 3/3 to Reviewer PyaS**
> > >
> > > > - **Regarding the quality of unconditional generation:** I do not quite grasp how the diffusion time step predictor improve the unconditional generation quality. Moreover, the experiments also lack strong and advanced baselines [3,4,5].
> > >
> > > Regarding the improved unconditional generation with the time step prediction, we answered the question in the general answer 2 above, because it was asked by most of the reviewers and we realized that we need to elaborate on it in more detail. Hence, we kindly ask you to refer to the general answer 2 above, where we hope we will be able to clearly address your concerns.
> > >
> > > Regarding the “lack of strong and advanced baselines”, we want to clarify that the cited work is not providing baselines for the experiments that we conduct. We sample atom positions from scratch given a composition of atoms as input. [5], on the other hand, generates both atom positions and types, which is different from our setup. Nevertheless, we have added as Table A1  the results by [5] and other established generative models that sample both positions and types.  We elaborate the different experimental setups in Appendix A.4.5. [3, 4] have been proposed for conformer search, the task of sampling diverse conformations given a molecular graph as input. It is related (and already included in our related work section) but a distinct task with different experimental setup and metrics for evaluation.
> > >
> > > We would like to note that the experiments in Section 4.3 were intended to evaluate  how much adaptive schedulingwith our time step predictor improves the traditional fixed schedule under the same condition, and not to demonstrate state-of-the-art performance. However, we thank the reviewer for pointing us into this direction, as evaluating the time step predictor in a state-of-the-art diffusion model for conformer search is an interesting additional experiment that we will explore. Unfortunately, it was infeasible for us to conduct this new kind of experiment (with new data and code base for the model) in the short time span of the rebuttal period.
> > >
> > > **Citations:**
> > >
> > > [6] Stocker, Sina, et al. "How robust are modern graph neural network potentials in long and hot molecular dynamics simulations?." Machine Learning: Science and Technology 3.4 (2022): 045010.
> > >
> > > [7] Frank, Thorben, Oliver Unke, and Klaus-Robert Müller. "So3krates: Equivariant attention for interactions on arbitrary length-scales in molecular systems." Advances in Neural Information Processing Systems 35 (2022): 29400-29413.
> > >
> > > [8] Unke, Oliver T., Stefan Chmiela, Michael Gastegger, Kristof T. Schütt, Huziel E. Sauceda, and Klaus-Robert Müller. "SpookyNet: Learning force fields with electronic degrees of freedom and nonlocal effects." Nature communications 12, no. 1 (2021): 7273.
> > >
> > > [9] Schütt, Kristof T., et al. "Schnet–a deep learning architecture for molecules and materials." The Journal of Chemical Physics 148.24 (2018).
> > >
> > > [10] Schütt, Kristof, Oliver Unke, and Michael Gastegger. "Equivariant message passing for the prediction of tensorial properties and molecular spectra." International Conference on Machine Learning. PMLR, 2021.
> > >
> > > [11] Satorras, Vıctor Garcia, Emiel Hoogeboom, and Max Welling. "E (n) equivariant graph neural networks." International conference on machine learning. PMLR, 2021.
> > >
> > > [12] Hoogeboom, Emiel, et al. "Equivariant diffusion for molecule generation in 3d." International conference on machine learning. PMLR, 2022.
> > >
> > > [13] Gasteiger, Johannes, Janek Groß, and Stephan Günnemann. "Directional message passing for molecular graphs." arXiv preprint arXiv:2003.03123 (2020).
> > >
> > > [14] Unke, Oliver T., and Markus Meuwly. "PhysNet: A neural network for predicting energies, forces, dipole moments, and partial charges." Journal of chemical theory and computation 15.6 (2019): 3678-3693.
> > >
> > > [15] Hoja, Johannes, et al. "QM7-X, a comprehensive dataset of quantum-mechanical properties spanning the chemical space of small organic molecules." Scientific data 8.1 (2021): 43.
> > >
> > > [16] Schütt, Kristof T., Stefaan SP Hessmann, Niklas WA Gebauer, Jonas Lederer, and Michael Gastegger. "SchNetPack 2.0: A neural network toolbox for atomistic machine learning." The Journal of Chemical Physics 158, no. 14 (2023).
> > >
> > > [17] Yang, Mengjiao, et al. "Scalable Diffusion for Materials Generation." arXiv preprint arXiv:2311.09235 (2023).

---

### Author Response · Authors · 2023-11-22
**General Answer 1.1 to All Reviewers**

Dear reviewers,

Thank you for taking your precious time to provide us with valuable feedback! We are glad that you found our submission:
- is a creative and novel approach to molecule relaxation
- includes a novel trick of time step prediction for reverse diffusion from non-Gaussian input
- is of great interest to chemistry, biology and other scientific communities
- is well presented and easy-to-understand in an intuitive manner with a well made connection to diffusion modeling and a great background on diffusion modeling
- has significant experimental results and the advantage of data efficiency in contrast to previous methods, which is significant because the training data collection is computationally very expensive.

Furthermore, your concerns have helped us to improve our submission and we believe that the revised manuscript provides a sound and significant contribution to the machine learning community. In the following, we will answer common questions and summarize a few changes that we made due to shared concerns of the reviewers. Additionally, we will provide individual point-by-point answers to each reviewer with more detailed elaborations. We would be very happy to see an increase in your rating if you feel that your concerns have been well addressed.


- Some reviewers have asked about the specific application domains of our approach, which made us realize that we need to address the scope of our contribution more clearly. We have made adaptations throughout the manuscript to achieve this and want to summarize our thoughts here:
Our main target application is to find stable molecule structures by relaxing unstable structures proposed by cheap methods, e.g., random conformation sampling and crude conformation optimization with empirical molecular mechanics force fields. For this application, machine learning force fields (FFs) are a well established and sound class of models that learn potential energy surfaces from labeled data, enabling fast yet accurate simulations and predictions that are not feasible with numerical quantum mechanics methods (DFT, CCSD(T)). We do not make the assumption or aim to replace FFs but to complement them. There exist many datasets such as MaterialsProject for crystal structure search, where only stable materials are reported. It is of great interest to identify novel stable materials but one cannot train FFs on these datasets due to a lack of unstable structures with force labels. Computing such training data is infeasible at high accuracy and for large systems. With MoreRed, we propose a novel approach that requires only stable molecules (without force label) for training. It learns to map unstable structures onto the manifold of stable structures using a pseudo energy landscape based on diffusion models, and with this we aim at potentially extending the applicability of ML methods to domains of structure search that are out of reach for FFs. Although our method is restricted to relaxation and cannot be used for e.g. molecular dynamics simulations like FFs, we believe that the potential applications are highly relevant and impactful.
For a fair comparison with the FF baseline, we used  the QM7-X dataset, which contains stable molecules of various compositions as well as corresponding unstable structures with force labels, and showed that MoreRed compares favorably to FF even though it only uses a small fraction (i.e., only stable molecules) of the training data. Evaluations on other systems such as materials remain for future work, as extending diffusion models to such structures with periodic boundaries is still an open research question in itself. However, as our framework is general in its choice of diffusion model, its applicability will expand with the advancements in diffusion models for atomistic systems, which, indeed, has gained significant momentum recently and is progressing rapidly.

---

> ### Author Response · Authors · 2023-11-22
> **General Answer 1.2 to All Reviewers**
>
> - The main methodological contribution of our work is the diffusion time step prediction and we have added more plots (Fig. 2) and analysis (Section 4.1) in the revision to stress and explain this point. It is an integral part of the MoreRed relaxation procedure.
> Diffusion models require a time step $t$ as input that depends on the noise level of the data point. Usually, one starts from complete Gaussian noise and the maximum $t=T$. Then $t$ is decreased by one each step and the denoising runs until $t=0$. When utilizing diffusion models for relaxation, however, one starts from an unstable structure that is far from the complete Gaussian noise. To this end, we require a mechanism that automatically determines the starting point of the relaxation. We solve this problem with the time step prediction. Our analysis and theoretical argument shows that the time step prediction is feasible (Fig. A1 which was  Fig.2 in the original submission) and that the distance of unstable structures to their stable counterpart clearly correlates with the time step prediction (Fig. 2a).  Since the unstable structure is not Gaussian distributed, the observation in Fig. 2a implies that the diffusion time predictor correctly measures the distance from the manifold of stable molecules, regardless of the type of noise.
> - Most reviewers asked for a more elaborate explanation of the improved unconditional generation with the time step prediction. We address this concern in the ‘General Answer 2 to All Reviewers’ for all reviewers, where we give a comprehensive and detailed explanation to clarify any ambiguity.
> - As asked by two reviewers, in order to validate the generality of our approach with respect to the choice of the equivariant backbone neural network, we have conducted experiments using the SO3net architecture as implemented in SchNetPack [1] - a spherical harmonics based neural network in the spirit of Tensor Field Networks [2] and NequIP [3] - in addition to the original experiments with PaiNN. The findings are reported in Appendix A.4.4 and Fig. A7, where we observe the same trend as with PaiNN: MoreRed outperforms the baseline FF in our study on QM7-X.
>
> To make it easier for the reviewers, we have marked our changes in the updated manuscript with blue color. Due to the page limit constraints, we have moved the old Fig. 2 to the appendix (now Fig. A1) since we believe that the new Fig. 2 contains more intuitive and relevant evaluations of the time step prediction for our application of molecule relaxation. Besides, we fixed some typos in the manuscript and a small typo in Equation 6, for which we also added the derivation in Appendix A.2.1.
> For more details, please see the separate answers to each reviewer. Thanks again for your precious feedback and helping us improve this submission.
>
> Kind regards,
>
>
> the authors
>
> **Citations:**
>
> [1] Schütt, Kristof T., et al. "SchNetPack 2.0: A neural network toolbox for atomistic machine learning." The Journal of Chemical Physics 158.14 (2023).
>
> [2] Thomas, Nathaniel, et al. "Tensor field networks: Rotation-and translation-equivariant neural networks for 3d point clouds." arXiv preprint arXiv:1802.08219 (2018).
>
>
> [3] Batzner, Simon, et al. "E (3)-equivariant graph neural networks for data-efficient and accurate interatomic potentials." Nature communications 13.1 (2022): 2453.

---

### Author Response · Authors · 2023-11-22
**General Answer 2 to All Reviewers: Improved Unconditional Generation with Time Step Prediction**

In addition to enabling denoising from arbitrary noise levels, the time step predictor is also a promising contribution to the field of diffusion models in its own regard and can potentially improve unconditional generation, as originally illustrated with our experiments in Section 4.3 and can be seen in the example samples in Appendix A.4.7. However, we realized that this was not explained clearly enough. Hence, we apologize for the ambiguity and to remedy these shortcomings we have updated the section about unconditional generation to better reflect this and expanded upon that in the Appendix A4.5 and added Fig. 2b and Fig. A8 (left) with examples of the trajectories of the predicted time steps through the denoising iterations for both sampling from complete noise and relaxation. Moreover, we summarize the details in the following:


First of all, we note that the advantage of the adaptive scheduling with the diffusion time prediction is already observed in the relaxation experiment in Fig. 3a, where MoreRed-AS/-JT, which adopt the adaptive scheduling, outperform MoreRed-ITP with the fixed scheduling.  The new Fig. 2b illustrates why the adaptive scheduling is advantageous.  The figure shows, for three unstable molecules as the starting points for relaxation, the trajectories of the adaptive diffusion time step (solid), the fixed diffusion step (dashed), and the predicted diffusion time when the fixed scheduling is applied (dotted) as a function of the number of reverse diffusion iterations.  When the fixed scheduling (dashed) is applied, the distance to the stable molecule manifold, which is implicitly measured as the predicted diffusion step (dotted), can be different from the scheduled noise level.  This mismatch is caused by the errors of the noise predictor, $\epsilon_{\theta}$ in Equation 7, accumulated over the reverse diffusion iterations. Consequently, the reverse diffusion can finish at $t = 0$ with the sample at which the diffusion time predictor still indicates that the sample has not reached the manifold of stable molecules. The adaptive scheduling on the other hand adjusts the diffusion time step (solid), according to the distance to the manifold measured by the time predictor, and takes as many steps as required to converge to the data manifold (Fig. 2b). This idea holds for diffusion models in general, independent of the starting point of denoising, and therefore can benefit unconditional data generation as well, as reported in Section 4.3. Previous work [1] has revealed the problems of the fixed reverse process of diffusion models, which can potentially lead to an erroneous sample $x_t$ at time step $t$ that deviates from the optimal reverse trajectory because of using one noise prediction per time step $t$ . If this error is not corrected it may propagate to later denoising iterations and get amplified, resulting in a worse sample quality. [1,2] partially mitigate this issue by performing correction steps after each diffusion reverse steps, e.g. by using second order SDE/ODE solvers or by running some Langevin dynamics iterations after each reverse step $t$. It was shown that these approaches can significantly improve the sample quality and that using more MCMC Langevin iterations during sampling could improve the results, while requiring additional  hyperparameter tuning, e.g., the number of correction steps per denoising iteration and the total number of denoising iterations. This is problematic because [2] showed that performing too many correction steps can hurt the quality of the samples, and that optimally the hyperparameters need to be tuned on each model and each dataset and we believe that this even extends to each sampling trajectory. We argue that our adaptive scheduling strategy is a promising solution to this issue, where correction steps are automatically added or reduced, according to the mismatch between the predicted diffusion time, which implicitly measures the distance from the data manifold, and the reverse diffusion iterations (as depicted in Fig. A8 left). Tables 1 and 2, where we compare two models that only differ in using the fixed schedule (DDPM) or the adaptive schedule (MoreRed), further support this hypothesis and showcase the significant improvement in unconditional molecular structure and image generation.

**Citations:**

[1] Yang Song, , Jascha Sohl-Dickstein, Diederik P Kingma, Abhishek Kumar, Stefano Ermon, Ben Poole. "Score-Based Generative Modeling through Stochastic Differential Equations." International Conference on Learning Representations. 2021.


[2] Tero Karras, , Miika Aittala, Timo Aila, Samuli Laine. "Elucidating the Design Space of Diffusion-Based Generative Models." Advances in Neural Information Processing Systems. 2022.

---

### Meta-Review · Area_Chair_puYR · 2023-12-08

**Metareview:**

The idea of this paper is novel: Generate stable molecular structures solely by using a stable molecule conformations. The learning problems is fomulated as a diffusion problem where the diffusion (noising process) is creating the unstable conformations and the denoising function is learned such that it has the stable conformation as target distribution.

It is unclear that the approach is in general well-founded, i.e. can learn meaningful stable conformation for general molecules. It hinges on that the diffused structures are sampling structures covers the relevant space of unstable structures around the stable structure. So the success will probably dependent quite a lot of the design of the diffusion process.

The reviewers are not convinced that the methodology in the paper is fully developed. Rejection is recommended with a strong encouragement to the authors to further develop this novel approach.

**Justification For Why Not Higher Score:**

Not quite ready for publication yet.

**Justification For Why Not Lower Score:**

None.

---

### Decision · Program_Chairs · 2024-01-16

Reject